# Cytotoxic CD8⁺ T cells promote granzyme B-dependent adverse post-ischemic cardiac remodeling

Icia Santos-Zas [1], Jeremie Lemarié[1,15], Ivana Zlatanova[1,15], Marine Cachanado[2], Jean-Christophe Seghezzi[3], Hakim Benamer[4], Pascal Goube[5], Marie Vandestienne[1], Raphael Cohen[1], Maya Ezzo[1], Vincent Duval[1], Yujiao Zhang[1], Jin-Bo Su [6], Alain Bizé[6], Lucien Sambin[6], Philippe Bonnin [7], Maxime Branchereau[8], Christophe Heymes[8], Corinne Tanchot[1], José Vilar [1], Clement Delacroix[1], Jean-Sebastien Hulot[1], Clement Cochain[9], Patrick Bruneval[1,10], Nicolas Danchin[11], Alain Tedgui [1], Ziad Mallat[1,12], Tabassome Simon[2,13], Bijan Ghaleh [6], Jean-Sébastien Silvestre [1] & Hafid Ait-Oufella [1,14 ✉]

Acute myocardial infarction is a common condition responsible for heart failure and sudden death. Here, we show that following acute myocardial infarction in mice, CD8⁺ T lymphocytes are recruited and activated in the ischemic heart tissue and release Granzyme B, leading to cardiomyocyte apoptosis, adverse ventricular remodeling and deterioration of myocardial function. Depletion of CD8⁺ T lymphocytes decreases apoptosis within the ischemic myocardium, hampers inflammatory response, limits myocardial injury and improves heart function. These effects are recapitulated in mice with *Granzyme B*-deficient CD8⁺ T cells. The protective effect of CD8 depletion on heart function is confirmed by using a model of ischemia/reperfusion in pigs. Finally, we reveal that elevated circulating levels of GRANZYME B in patients with acute myocardial infarction predict increased risk of death at 1-year follow-up. Our work unravels a deleterious role of CD8⁺ T lymphocytes following acute ischemia, and suggests potential therapeutic strategies targeting pathogenic CD8⁺ T lymphocytes in the setting of acute myocardial infarction.

[1] Université de Paris, PARCC, INSERM, F-75015 Paris, France. [2] Assistance Publique-Hôpitaux de Paris, APHP.SU; Department of Clinical Pharmacology and Clinical Research Platform (URCEST-CRB-CRC-EST), Hôpital Saint Antoine, Paris, France. [3] Service de cardiologie, Centre Hospitalier de Compiegne, Compiegne, France. [4] Service de cardiologie, Institut Cardiovasculaire Paris Sud, Paris, France. [5] Service de cardiologie, Centre Hospitalier de Corbeil, Corbeil, France. [6] Inserm U955-IMRB, Equipe 03, UPEC, Ecole Nationale Vétérinaire d'Alfort, Maisons-Alfort, France. [7] Inserm U965, Department of Physiology, Assistance Publique Hôpitaux de Paris, Hôpital Lariboisière, France. [8] Inserm U1048-Institut des Maladies Métaboliques et Cardiovasculaires (I2MC), université Paul Sabatier, Toulouse, France. [9] Institute of Experimental Biomedicine, University Hospital Würzburg, Würzburg, Germany. [10] Service d'anatomopathologie, Hôpital Europeen G. Pompidou, Assistance Publique, Hôpitaux de Paris, Paris, France. [11] Service de cardiologie, Hôpital Europeen G. Pompidou, Assistance Publique, Hôpitaux de Paris, Paris, France. [12] Division of Cardiovascular Medicine, University of Cambridge, Addenbrooke's Hospital, Cambridge CB2 2QQ, UK. [13] Sorbonne Université, UPMC-site St Antoine, Service de Pharmacologie, Assistance Publique-Hôpitaux de Paris, APHP.SU; Department of Clinical Pharmacology and Clinical Research Platform (URCEST-CRB-CRC-EST), Hôpital Saint Antoine, Paris, France. [14] Sorbonne Université, Service de médecine intensive-Réanimation, Assistance Publique, Hôpitaux de Paris, Paris, France. [15]These authors contributed equally: Jeremie Lemarié, Ivana Zlatanova. ✉email: hafid.aitoufella@inserm.fr

Acute coronary syndrome including myocardial infarction (MI) is the most prevalent manifestation of cardiovascular diseases and is associated with high mortality and morbidity. Nevertheless, considerable advances have been achieved in the early management of acute coronary thrombotic occlusion, including rapid mechanical restoration of coronary artery blood flow and anti-platelet therapies[1]. Consequently, early mortality of patients with MI has declined over the last decades in the United States[2] and Europe[3]. However, long-term effects of ischemia-related cardiac damage continue to be a clinical and social burden, due to increased risk of arrhythmias, heart failure, and repetitive hospitalizations[4]. Therefore, more efforts have to be deployed toward the development of therapeutic approaches targeting pathophysiological pathways involved in post-ischemic cardiac remodeling.

There is a large body of human and experimental evidence showing that the immune response is involved in the long-term cardiac complications of coronary occlusion[5]. In human and experimental MI, interruption of blood supply leads to rapid death of cardiac myocytes in the ischemic heart. Thereafter, inflammatory signals trigger the recruitment and proliferation of immune/inflammatory cells, which contribute to left ventricle (LV) remodeling, at least in part, by modulating the production and degradation of extracellular matrix proteins, as well as the clearance of dead cardiac myocytes and their debris. Among several immune cell types that populate the infarcted myocardium, CD4+ T cells have been shown to infiltrate heart tissue within the first week following acute MI[6]. Re-supplementation experiments showed that CD4+ T cells contribute to myocardial ischemia-reperfusion injury through IFN-γ production. Conversely, natural regulatory T cells (Tregs) have been found to protect against deleterious inflammatory remodeling following MI[7,8]. Treg depletion using an anti-CD25 antibody impaired left ventricular dilation and survival, whereas expanding Tregs in vivo attenuated myocardial pro-inflammatory cytokine expression and leukocyte recruitment[7,8]. T cell receptor (TCR)-independent[9] and -dependent mechanisms[10] are involved in CD4+ T cell-related effects following myocardial ischemia-reperfusion.

However, the role of cytotoxic CD8+ T lymphocytes, another subset of T cells that orchestrate immune responses against viruses and tumors, remains unclear in the context of acute heart ischemia. Although CD8+ T cells have been detected in the heart following coronary artery occlusion[11] in rodents, some authors suggested that CD8+ T cells had no pathogenic role[12], whereas others have reported that a subset of CD8+ T cells expressing the type 2 angiotensin II receptor may be protective[13]. In a human observational study, an acute reduction in blood CD8+ T lymphocyte count has been reported within 1 h after coronary artery reperfusion, the drop of CD8+ T cells being more important in patients who developed heart microvascular obstruction[14].

Here, we showed that, following acute MI in mice, CD8+ T lymphocytes are recruited in the ischemic heart and foster cardiomyocyte death through the local release of Granzyme B, leading to enhanced myocardial inflammation, tissue injury, and deterioration of myocardial function. We also unraveled that immunotherapy based on the administration of neutralizing antibody directed against CD8+ T cells promotes cardiac repair in a mouse model of MI as well as in a pig model of cardiac ischemia-reperfusion.

## Results

### Cytotoxic CD8+ T lymphocytes are recruited and activated in the ischemic heart tissue after MI.
First, we characterized the kinetics of inflammatory cell recruitment within the injured myocardium. We used an experimental model of acute MI in C57BL/6J mice induced by permanent coronary artery ligation and we analyzed cell suspensions of digested hearts at different time points after the onset of ischemia, by flow cytometry. We found that CD3+CD8+ T (Fig. 1A–C) and CD3+CD4+ T lymphocytes accumulated in the injured myocardium as early as day 1 and peaked at day 3 after MI. Infiltration of CD4+ T cell was 2-fold higher than that of CD8+ T cells (Fig. 1D). Immunohistological analyses confirmed the increased accumulation of CD3+CD8+ T lymphocytes in both peri-infarct and infarct areas after MI compared to sham-operated animals (Fig. 1C and Supplementary Figs. 1–3). At day 1, infiltrating CD8+ T cells were mostly naive and the proportion of CD44+CCR7high central memory and CD44+CCR7low effector memory CD8+ T cells increased over time (Fig. 1E and Supplementary Fig. 4). Flow cytometry analyses showed that the proportion of CD8+ T cells expressing CD69 and CD107a increased during the first week after MI in the heart (Fig. 1F), as well as in the draining mediastinal lymph nodes (Supplementary Fig. 5). Local activation and degranulation of recruited cytotoxic CD8+ T cells were confirmed by the detection of Granzyme B within the ischemic heart tissue at day 1 after MI at both protein (Fig. 1G) and mRNA levels (Fig. 1H). Immunofluorescence staining showed colocalization of Granzyme B and CD8+ T cells. Of note, Granzyme B was mainly detected in the cytoplasm of T cells in non-ischemic areas, and around CD8+ T cells in ischemic areas, suggesting active CD8+ T cell degranulation in the infarcted myocardium (Fig. 1G and Supplementary Fig. 6).

Previous studies suggest that CD4+ T cells may orchestrate myeloid and lymphoid cell recruitment[15,16]. We quantified T subsets at day 1 and day 3 after MI in mice treated with anti-CD4 depleting monoclonal antibody or isotype control. At day 1, CD8+ T cell number was decreased in blood (Supplementary Fig. 7a), but increased in spleen after CD4+ T cell depletion (Supplementary Fig. 7b, c). At day 3, we observed that CD4+ T cell depletion (Fig. 1I) led to a 50% reduction in infiltrating CD8+ T cells in heart tissue (P < 0.01) (Fig. 1J, K), indicating that CD4+ T cells partly controlled CD8+ T cell mobilization from spleen into peripheral blood and ultimately their infiltration into the infarcted heart.

These findings indicate that circulating CD8+ T cells are recruited into the myocardium following MI, are activated and release Granzyme B, suggesting a potential role of CD8+ T cell-mediated immune response in this setting.

### CD8+ T lymphocyte depletion prevents adverse ventricular remodeling and improves cardiac function after acute MI in mice.
To directly assess the role of CD8+ T lymphocytes in cardiac remodeling after acute MI, we depleted CD8+ T lymphocytes using a CD8-specific monoclonal antibody (CD8 mAb)[17] 1 h after coronary ligation. CD8 mAb treatment rapidly depleted CD8+ T cells (>98%) within 6 h after mAb injection (Supplementary Fig. 8) in the peripheral blood (Fig. 2A and Supplementary Fig. 9), the cardiac tissue (Fig. 2B), and the spleen (Supplementary Fig. 10). Complete CD8+ T cell depletion was confirmed by immunofluorescence in ischemic heart tissue (Fig. 2C and Supplementary Fig. 8). CD8+ T cell depletion did not significantly impact survival following MI (at day 21, 76% in control group versus 92% in CD8-depleted group, P = 0.28). We next assessed cardiac function by echocardiography 9 and 28 days after MI. CD8 mAb-induced T cell depletion led to smaller end-systolic (P < 0.01) and end-diastolic left ventricular volumes (P < 0.05) (Fig. 2D), and to a significant improvement in left ventricular fractional shortening (Fig. 2D and Supplementary Fig. 11)

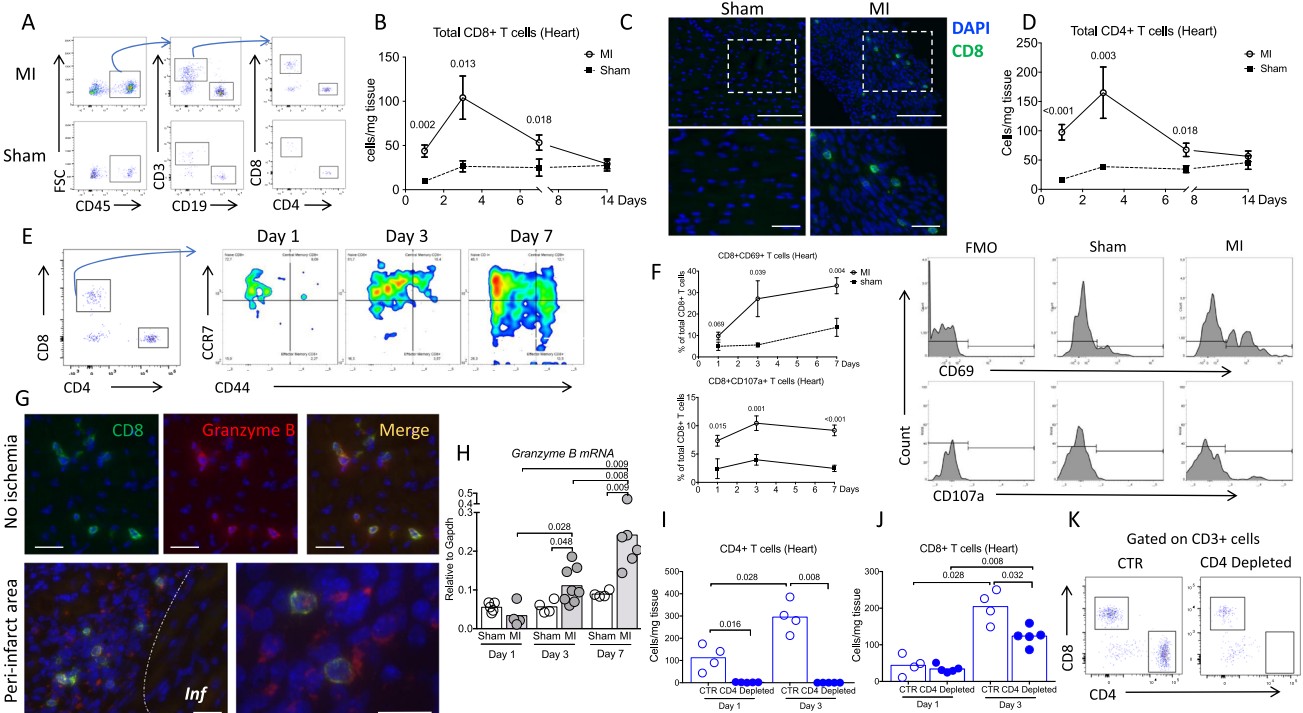

**Fig. 1 Cytotoxic CD8+ T lymphocytes are activated and recruited to the ischemic tissue after myocardial infarction. A** Representative examples of CD4+ and CD8+ T cell staining in the heart of C57BL/6J mice following coronary ligation (MI) or sham surgery. **B** Kinetic of CD8+ T cell infiltration in the myocardium, performed at days 0, 1, 3, 7, and 14 after surgery in MI (white circle, n = 9/12/12/5, respectively, at day 1/3/7/14) and sham-operated mice (black box, n = 6/time point); data are presented as mean ± SEM. **C** Immunostaining in the ischemic myocardium at day 3 after MI or sham showing CD8+ T infiltration (green) in the ischemic heart tissue, scale bar 40 μm. **D** Kinetic of CD4+ T cell infiltration in the myocardium, performed at days 0, 1, 3, 7, and 14 after surgery in MI (white circle, n = 10/9/12/5, respectively, at day 1/3/7/14) and sham-operated mice (black box, n = 6/time point); data are presented as mean ± SEM. **E** Flow cytometry characterization of CD8+ T cell subsets in the ischemic heart at days 1, 3, and 7 after MI including naïve (CD3+CD8+CCR7HighCD44−), effector memory (CD3+CD8+CCR7LowCD44+), and central memory (CD3+CD8+CCR7HighCD44+) subsets. **F** Representative examples and quantitative analysis of CD8+ T cells expressing CD69 or CD107a within the ischemic heart tissue in MI (white circle, n = 6/9/12, respectively, at day 1/3/7) and sham-operated mice (black box, n = 6/time point); data are presented as mean ± SEM. **G** Immunostaining in the ischemic myocardium at day 3 after MI showing CD8+ T cells (green), and Granzyme B (Red) and merged area (Yellow), Inf for infarct area; scale bar 40 μm. **H** mRNA levels of Granzyme B within the injured myocardium on days 1, 3, and 7 after coronary ligation (gray, n = 5/4/5 at day 1/3/7) or sham (white, n = 4/8/6 at day 1/3/7). **I** C57Bl6 WT mice received intraperitoneal injection of isotype control (CTR, blue borderline, n = 4/time point) or anti-CD4 depleting monoclonal antibody (blue filled, n = 5/time point) (150 μg/mice) 1 day before coronary occlusion. CD4+ T cell depletion was confirmed in the heart by flow cytometry at day 1 and day 3 following MI. **J**, **K** Quantification and representative example of CD8+ T cell count in the heart of control (n = 4/time point) or CD4-depleted (n = 5/time point) mice at day 1 and day 3 after MI. P values were calculated using two-tailed Mann-Whitney test (**B**, **D**, **F**, **H**, **I**, **J**). Inf, infarct; MI, myocardial infarction; FMO, fluorescent minus one.

compared to isotype-treated control mice at both time points. At day 28, left ventricular pressures were measured using an intra-cardiac probe, confirming that CD8 depletion preserved both diastolic and systolic LV functions (Fig. 2E). Of interest, left ventricular myocardial contractility was also improved by CD8 T cell depletion (P < 0.001) (Fig. 2F).

CD8 mAb-induced improvement in cardiac function was associated with abrogation of adverse LV remodeling. Infarct size (Fig. 2G and Supplementary Fig. 12) (P < 0.001) and interstitial fibrosis (P < 0.05) assessed by collagen content, were reduced in CD8 mAb-treated mice compared to isotype-treated control animals, at day 21 post-MI (Fig. 2H). The protective effect of CD8 T cell depletion was also confirmed on female C57BL/6 mice (Supplementary Fig. 13). CD8 T cell depletion decreased collagen synthesis as revealed by the reduction of *Col1a1* and *Col1a3* mRNA levels (Supplementary Fig. 14). Such protective effect of CD8 depletion was maintained at day 56 following MI (Supplementary Fig. 15). In summary, these results show that systemic CD8+ T cell depletion significantly reduces post-ischemic heart injury, prevents adverse ventricular remodeling, and improves cardiac function after acute MI.

**CD8+ T cells' pathogenic activity requires TCR engagement.** To assess the putative role of antigen recognition by CD8+ T cells, we used OT-I mice, in which the majority of CD8+ T cells exclusively recognize an irrelevant ovalbumin-derived peptide via their TCR. In a first set of experiments, coronary artery ligation was performed in male OT-I mice and 1 h later, animals were injected either with an isotype control or an anti-CD8-depleting antibody (Fig. 3A). In this setting, CD8 T cell depletion (Fig. 3B) did not impact infarct size at day 21 post-MI (Fig. 3C, D). To further substantiate the role of TCR-mediated pathogenic activity of CD8+ T cells, we injected *Rag1−/−* mice with CD8+ T cell-depleted splenocytes, re-supplemented with wild-type (WT) or OT-I CD8+ T lymphocytes (Fig. 3E). Survival at day 21 was not statistically different between groups despite a trend toward a better survival in OT-I CD8+ T cell-supplemented group (Fig. 3F). Animals re-supplemented with OT-I CD8+ T cells displayed less cardiac damage with a reduction in the infarct size (Fig. 3G) (P = 0.054) and a better cardiac function (Fig. 3H) (P < 0.01) than animals re-supplemented with WT CD8+ T cells.

Finally, we employed a third approach to address the importance of CD8+ T antigen-specific response using *CMy-mOva* mice.

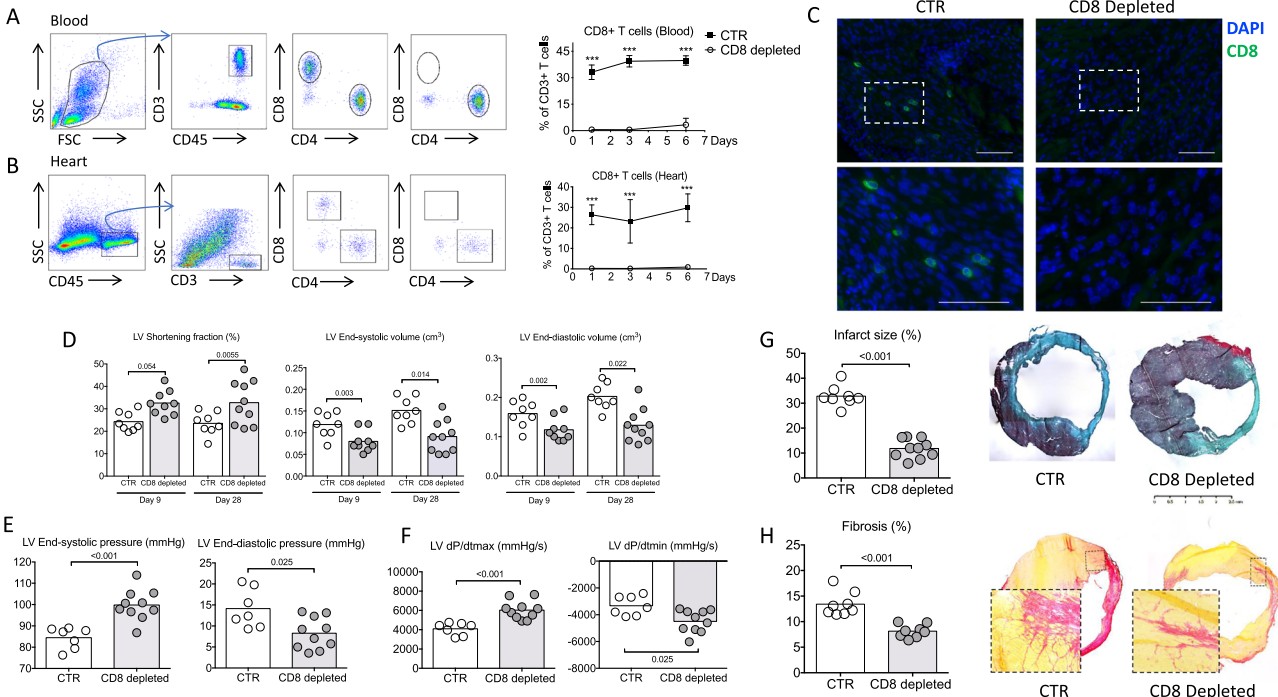

**Fig. 2 CD8 T cell depletion improves heart function and reduces infarct size. A** Representative examples (left) and quantitative analysis (right) of CD8+ T cell staining in the blood of C57BL/6J mice treated with isotype control (CTR, black box) or with the CD8 mAb (CD8 depleted, white circle) (n = 6 mice per group/time point); data are presented as mean ± SEM, ***P < 0.001. **B** Representative examples (left) and quantitative analysis (right) of CD8+ T cell staining in the heart of C57BL/6J mice treated with isotype control (CTR) or with the CD8 mAb (CD8 depleted) (n = 6 mice per group/time point); data are presented as mean ± SEM, ***P < 0.001. **C** Representative examples of CD8+ T cell staining in the peri-infarct area of C57BL/6J mice treated with isotype control (CTR) or with the CD8 mAb (CD8 depleted) at day 3, scale bar 40 μm. **D** Echocardiography analysis after anti-CD8 therapy. We measured LV shorting fraction (SF), LV volume at end systole and end diastole *(CTR white, n = 8 and CD8 depleted gray n = 9-10). **E** Systolic and diastolic pressure measured in the LV, and **F** related function parameters using intracardiac probe at day 28 (CTR n = 7 and CD8 depleted n = 9-10). **G** Representative photomicrographs and quantitative analysis of infarct size evaluation evaluated by Masson trichrome staining, in the 2 groups of mice (CTR n = 8 and CD8 depleted n = 10). **H** Representative photomicrographs and quantitative analysis of myocardial fibrosis evaluated by Sirius Red staining, in the 2 groups of mice (CTR n = 8 and CD8 depleted n = 9). LV, left ventricle. P values were calculated using two-tailed Mann-Whitney test (**A**, **B**, **D**, **E**, **F**, **G**, **H**).

*CMy-mOva* mice is a transgenic mouse line that expresses cardiac myocyte restricted membrane-bound ovalbumin[18] that can be recognized by OT-I CD8+ T cells. Three days before MI, *CMy-mOva* mice were injected either with WT or OT-I purified CD8+ T lymphocytes (Fig. 3I). The injection of OT-I CD8+ T lymphocytes enhanced *Granzyme B* mRNA content in the ischemic heart 2 days after MI when compared to control group (Fig. 3J). In addition, the injection of OT-I CD8+ T lymphocytes increased mortality rate (85% versus 40%, P < 0.01) (Fig. 3K) and infarct size among rare survivors (Fig. 3L) when compared to animals receiving WT CD8+ T cells.

**CD8+ T lymphocyte depletion reduces cardiomyocyte apoptosis and pro-inflammatory responses after acute MI.** We next assessed the potential mechanisms involved in CD8+ T cell-mediated effects on cardiac remodeling and function. We first assessed the effect of CD8+ T cell deficiency on other actors of the inflammatory reaction. As shown in Supplementary Figs. 16–21, CD8+ T cell depletion had no impact on the number of CD4+ T cells, B cells, NK, NKT, classical monocytes, neutrophils, macrophages, and dendritic cells in the ischemic heart tissue.

Cytotoxic activity of CD8+ T cells in the context of cancer[19] or virus infection[20] is mainly mediated by the release of Perforin/Granzyme B[21]. Of note, Granzyme B colocalized with apoptotic cells in the ischemic heart tissue (Supplementary Fig. 22). CD8+ T cell depletion reduced Granzyme B content in the ischemic

heart tissue at both mRNA (Fig. 4A) and protein levels (Fig. 4B). Such decrease in cardiac Granzyme B content was associated with a significant reduction of TUNEL+ apoptotic myocardial cells (Fig. 4C and Supplementary Fig. 23) and a diminution of infarct area 3 days after MI (Supplementary Fig. 24). Treatment with CD8 mAb also reduced local pro-inflammatory cytokine levels at day 7 after MI. Of note, *Tnf-α*, *Il-1β*, and *Il-6* mRNA levels were significantly lower (P < 0.05) in infarct hearts of CD8-depleted mice compared to the control group (Fig. 4D). In addition, we found a marked decrease of *Mmp9 gene* expression (Fig. 4E) and a substantially lower metalloproteinase activity (Fig. 4F) in the heart of mice treated with anti-CD8-depleting antibody. Such alteration in the inflammatory landscape without any difference in the number of infiltrating leukocyte subsets suggests a mAb CD8-mediated immune phenotypic switch toward an anti-inflammatory profile. As such, cardiac macrophages displayed a reparative anti-inflammatory signature as revealed by the reduction of *Il-1β*, *Tnf-α*, and *iNos* mRNA levels in macrophages of anti-CD8-treated mice (Supplementary Fig. 25). On the same note, the number of reparative macrophage expressing CD206 was increased in the heart of CD8-depleted animals at day 7 (Supplementary Fig. 26).

**Global Granzyme B deficiency limits cardiac damage after acute MI.** These findings prompted us to investigate the direct cytotoxic role of Granzyme B in post-ischemic cardiac

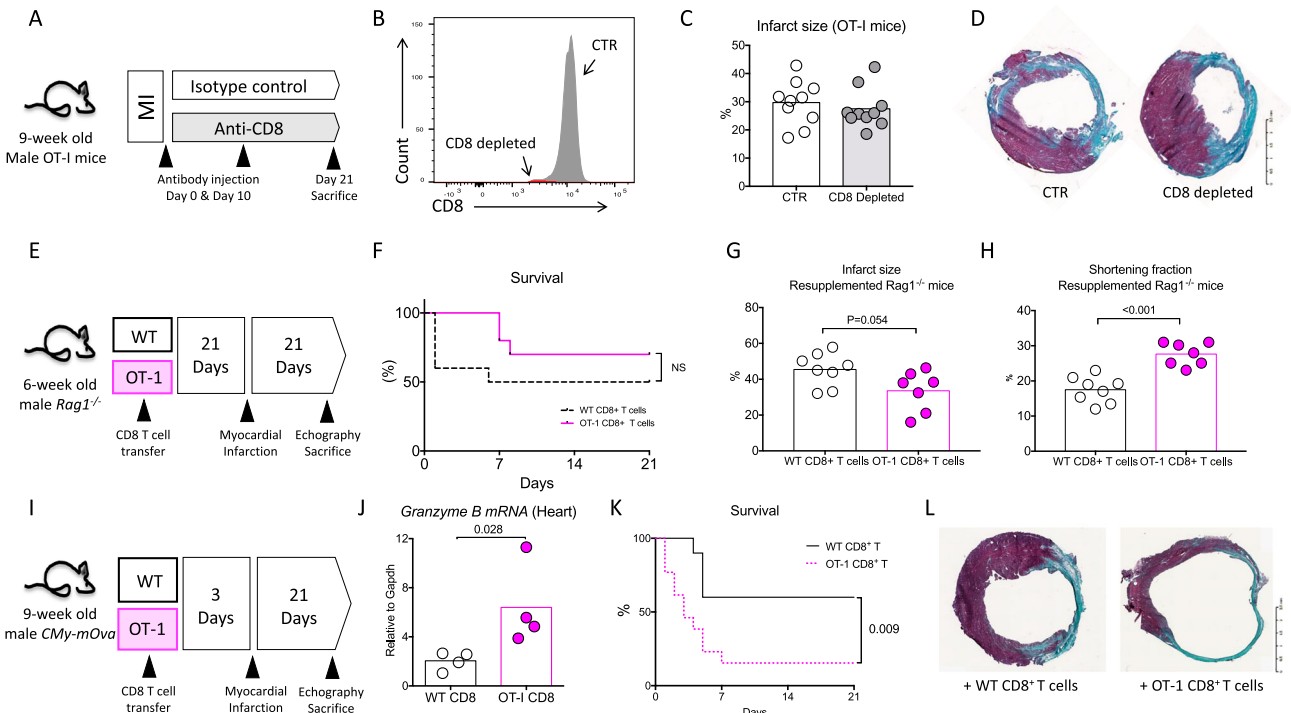

**Fig. 3 CD8$^+$ T cells pathogenic activity requires antigen-specific stimulation. A** Male 9-week-old OT-I mice were treated with isotype control (CTR, white) or the CD8 mAb (CD8 depleted, gray). **B** CD8 depletion (red) was confirmed in the spleen at day 21 using flow cytometry. **C**, **D** Representative photomicrographs and quantitative analysis of infarct size evaluation using Masson trichrome staining, in the 2 groups of OT-I mice (CTR $n = 10$ and CD8 depleted $n = 9$). **E** Rag1$^{-/-}$ mice injected with CD8-depleted splenocytes re-supplemented with WT (white) or OT-I (pink) CD8$^+$ T cells, 3 weeks before MI. **F** Survival rate following MI (from 2 experiments, WT $n = 16$ and OT-I $n = 13$). **G** Quantitative analysis of infarct size evaluation assessed by Masson trichrome staining in the 2 groups of mice (WT $n = 8$ and OT-I $n = 7$). **H** Echocardiography analysis 21 days after MI and assessment of LV shortening fraction (SF) in the 2 groups of re-supplemented mice (WT $n = 8$ and OT-I $n = 7$). **I** CMy-mOva mice were injected with WT (white) or OT-I (pink) CD8$^+$ T cells, 3 days before MI. **J** Granzyme B mRNA expression in the ischemic heart at day 2 after MI in CMy-mOva mice injected with WT or OT-I CD8$^+$ T cells ($n = 4$/group). **K** Survival rate following MI (pooled 2 experiments, WT $n = 10$ and OT-I $n = 12$). **L** Representative examples of infarct size after Masson trichrome staining. P values were calculated using two-tailed Mann-Whitney test (**C**, **G**, **H**, **J**). Difference in survival was evaluated using log-rank test (**F**, **K**).

remodeling. First, MI was induced in *C57bl6* WT and Granzyme B-deficient (*GzmB*$^{-/-}$) adult mice. Granzyme B deficiency was confirmed by immunostaining in the spleen of *GzmB*$^{-/-}$ mice as well as in the heart (Supplementary Fig. 27). Following acute MI, CD8$^+$ T cell infiltration was observed in the ischemic heart of *C57bl6* and *GzmB*$^{-/-}$ mice (Supplementary Fig. 28). A significant reduction of TUNEL+ apoptotic cells was found within the injured myocardium ($P < 0.001$) (Fig. 4G), as well as a local reduction of *Il-1β, Il-6, Tnf-α*, and *Mmp9* mRNA levels ($P < 0.01$) in *GzmB*$^{-/-}$ mice compared to WT control group (Fig. 4H and Supplementary Fig. 29). Finally, at day 21 following MI, the infarct size was markedly reduced in *Granzyme B* deficient animals ($-55\%$, $P < 0.05$) (Fig. 4I), and overall survival trended toward improvement (88% vs 64%, $P = 0.12$) (Supplementary Fig. 30). These experiments suggest that Granzyme B per se may have direct cytotoxic activity on cardiomyocytes. To test this hypothesis, mouse cardiomyocytes were co-cultured in vitro with purified WT or GzmB$^{-/-}$ splenic CD8$^+$ T cells. After 24 h, T cells were removed and cardiomyocyte apoptosis was monitored for an additional 24-h period using caspase-3 fluorescent dye. CD8$^+$ T cell activation was achieved using dynabeads mouse T activator CD3/CD28 (Supplementary Fig. 31). Pre-incubation with activated CD8$^+$ T cells did not increase cardiomyocyte apoptosis when compared to pre-incubation with non-activated CD8$^+$ T cells at low concentrations (Cardiomyocyte/CD8 ratio 1/1 and 1/3) (Fig. 4J). However, at higher concentrations (ratio 1/5 and 1/10), activated CD8$^+$ T cells strongly promoted

cardiomyocyte apoptosis measured as Caspase 3/7+ cells (Fig. 4J). Cytotoxicity of CD8$^+$ T cells was abolished in case of Granzyme B deficiency (Fig. 4K). We also hypothesized that the effect of CD8 T lymphocytes expressing Granzyme B exceeds a simple cytotoxic action and could impair cardiomyocyte function. For this purpose, cardiomyocytes isolated from adult C57Bl/6J mice were co-cultured overnight with purified WT or *GzmB*$^{-/-}$ spleen CD8$^+$ T cells at low concentrations, i.e., 1/3 ratio. Cardiomyocyte contractility was evaluated using computer-assisted sarcomere shortening measurements. Interestingly, decreased sarcomere shortening was observed in cardiomyocytes co-cultured with activated WT CD8$^+$ T cells, compared with cardiomyocytes co-cultured with control WT CD8$^+$ T cells or activated *GzmB*$^{-/-}$ CD8$^+$ T lymphocytes (Fig. 4L). Altogether, our results suggest that low number of CD8$^+$ T cells curb cardiomyocyte contractility, and that high number of CD8$^+$ T cells precipitates cardiomyocyte death.

**Granzyme B-deficient CD8$^+$ T lymphocytes fail to affect cardiac remodeling and function after acute MI.** To further substantiate the role of CD8$^+$ T cell-derived Granzyme B, we injected Rag1$^{-/-}$ mice either with CD8$^+$ T cell-depleted splenocytes, CD8$^+$ T-depleted splenocytes re-supplemented with WT or *GzmB*$^{-/-}$ CD8$^+$ T lymphocytes (Fig. 5A). The purity of the CD8$^+$ T lymphocytes is shown in Supplementary Fig. 32. We first verified that re-supplementation with WT or *GzmB*$^{-/-}$ CD8$^+$ T

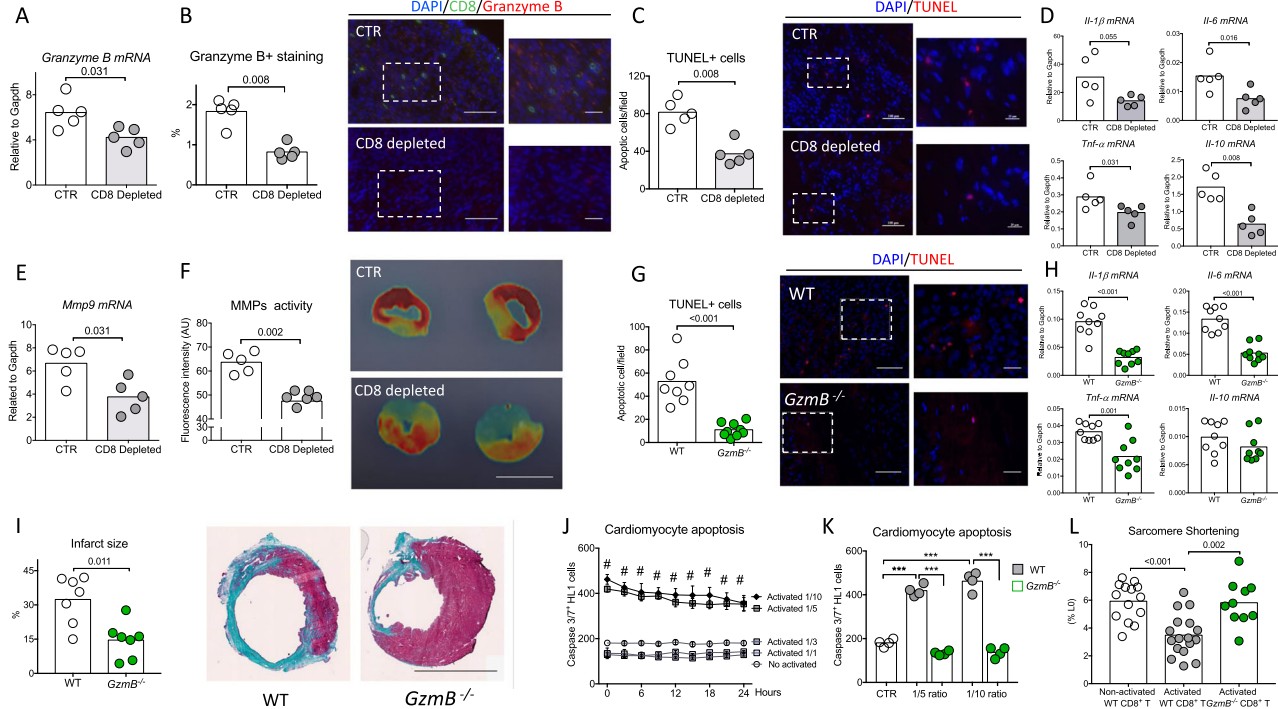

**Fig. 4 CD8+ T lymphocyte depletion or Granzyme B global deficiency reduces cardiomyocyte apoptosis and pro-inflammatory responses within the ischemic heart tissue. A** Representative histograms of mRNA levels of Granzyme B within the injured myocardium on day 3 after MI in CTR (white) and CD8-depleted (gray) mice ($n = 5$/group). **B** Representative examples (right) and quantitative analysis (left) of Granzyme B staining in the ischemic heart of C57BL/6J mice with or without CD8 depletion ($n = 5$/group); scale bars 50 and 25 μm. **C** Representative examples (right) and quantitative analysis (left) of TUNEL+ cells (Red) in the peri-infarct area of C57BL/6J mice ($n = 5$/group); scale bars 50 and 25 μm. **D** Representative histograms of mRNA levels of Il-1β, I-6, Tnf-α, and Il-10 within the injured myocardium on day 7 after MI ($n = 5$/group). **E** Representative histograms of mRNA levels of *Mmp9* within the injured myocardium on day 7 after MI ($n = 5$/group). **F** Quantification (left) and representative photomicrographs (right) of matrix metalloproteinase (MMP)-sense 680 activity in the ischemic heart measured by ex vivo reflectance epifluorescence imaging at day 7 (CTR $n = 5$ and CD8 depleted $n = 6$), scale bar 2.5 mm. **G** Acute MI was induced on *C57bl6* wild-type (WT, white) mice or *Granzyme B* deficient (*GzmB−/−*, green) mice. Representative examples (right) and quantitative analysis (left) of TUNEL+ cells in the peri-infarct area of WT C57BL/6J or *GzmB−/−* mice at day 3 after MI (WT $n = 8$ and *GzmB−/−* $n = 9$); scale bars 50 and 25 μm. **H** Il-1β, Il-6, Tnf-α, and Il-10 mRNA levels measured by qPCR in infarcted heart at day 3 after MI ($n = 9$/group). **I** Representative photomicrographs (left) and quantitative analysis (right) of infarct size evaluation using Masson trichrome staining, in the 2 groups of mice ($n = 7$/group); scale bar 2.5 mm. **J** Purified non-activated or activated WT CD8+ T cells were co-cultured with cardiomyocytes at different ratios (Cardiomyocyte/CD8+ T cells) for 24 h before their removal. Apoptotic cardiomyocytes labeled with an active caspase-3 fluorescent dye was monitored for 24 h ($n = 4$–5/conditions), Data are presented as mean ± SEM, #$P < 0.001$ for activated CD8+ T cells 1/5 versus non-activated CD8+ T cells or activated CD8+ T cells 1/1 or activated CD8+ T cells 1/3; #$P < 0.001$ for activated CD8+ T cells 1/10 versus non-activated CD8+ T cells or activated CD8+ T cells 1/1 or activated CD8+ T cells 1/3. **K** WT or *GzmB−/−* CD8+ T cells were co-cultured with cardiomyocytes for 24 h at 1/5 and 1/10 ratio and cardiomyocyte apoptosis using an active caspase-3 fluorescent dye was quantified. Cardiomyocyte and non-activated CD8+ T cells co-culture was named CTR condition ($n = 4$/condition) ***$P < 0.001$. **L** Isolated cardiomyocytes were co-cultured overnight with CD8+ T cells isolated from WT or *GzmB−/−* mice and cardiomyocyte sarcomere shortening was measured (non-activated $n = 14$, activated $n = 17$ and *GzmB−/−* $n = 10$) at a ratio 1/3. *P* values were calculated using two-tailed Mann-Whitney test (**A, B, C, D, E, F, G, H, I, J**) or Kruskal-Wallis test (**K, L**). TUNEL, terminal deoxynucleotidyl transferase dUTP Nick End Labeling; MMP, matrix metalloprotease.

lymphocytes significantly increased CD8+ T cell numbers in spleens and hearts of *Rag1−/−* mice compared to mice injected with CD8+ T cell-depleted splenocytes only (Supplementary Fig. 33).

We then examined the consequences of Granzyme B deficiency in CD8+ T lymphocytes on post-ischemic cardiac remodeling. Transfer of WT CD8+ T cells into *Rag1−/−* mice reduced survival (Fig. 5A, B) and left ventricular shortening fraction (Fig. 5E) ($P < 0.05$) after MI compared to the transfer of CD8-depleted splenocytes. In our re-supplementation experiment, we observed a significant negative correlation between WT CD8+ T cell number and cardiac function (Fig. 5F). This pathogenic effect on mortality and LV systolic function was abrogated after re-supplementation with *GzmB−/−* CD8+ T lymphocytes (Fig. 5B–E). CD8+ T cell supplementation also increased infarct size ($P = 0.04$; Fig. 5C) and collagen content (Fig. 5D), which was

prevented by re-supplementation with *GzmB−/−* CD8+ T lymphocytes (Fig. 5C, D).

## CD8+ T lymphocyte depletion is protective in a pig model of coronary ischemia/reperfusion.

To confirm the pathogenic role of CD8+ T cells in MI and to substantiate the therapeutic interest of CD8-depleting antibody, we used a model of cardiac ischemia-reperfusion in pigs. To achieve CD8+ T depletion, we used an IgG2a mouse anti-swine monoclonal anti-CD8 antibody (clone 76-2-11) with known in vitro[22] and in vivo activity against porcine CD8+ T cells[23,24]. As shown in supplementary Fig. 34, pig anti-CD8 mAb treatment was efficient but induced delayed CD8+ T cell depletion when compared to mouse anti-CD8 mAb. Indeed, full CD8 depletion was obtained at day 3 after pig anti-CD8 mAb injection, whereas complete CD8+ T cell depletion was

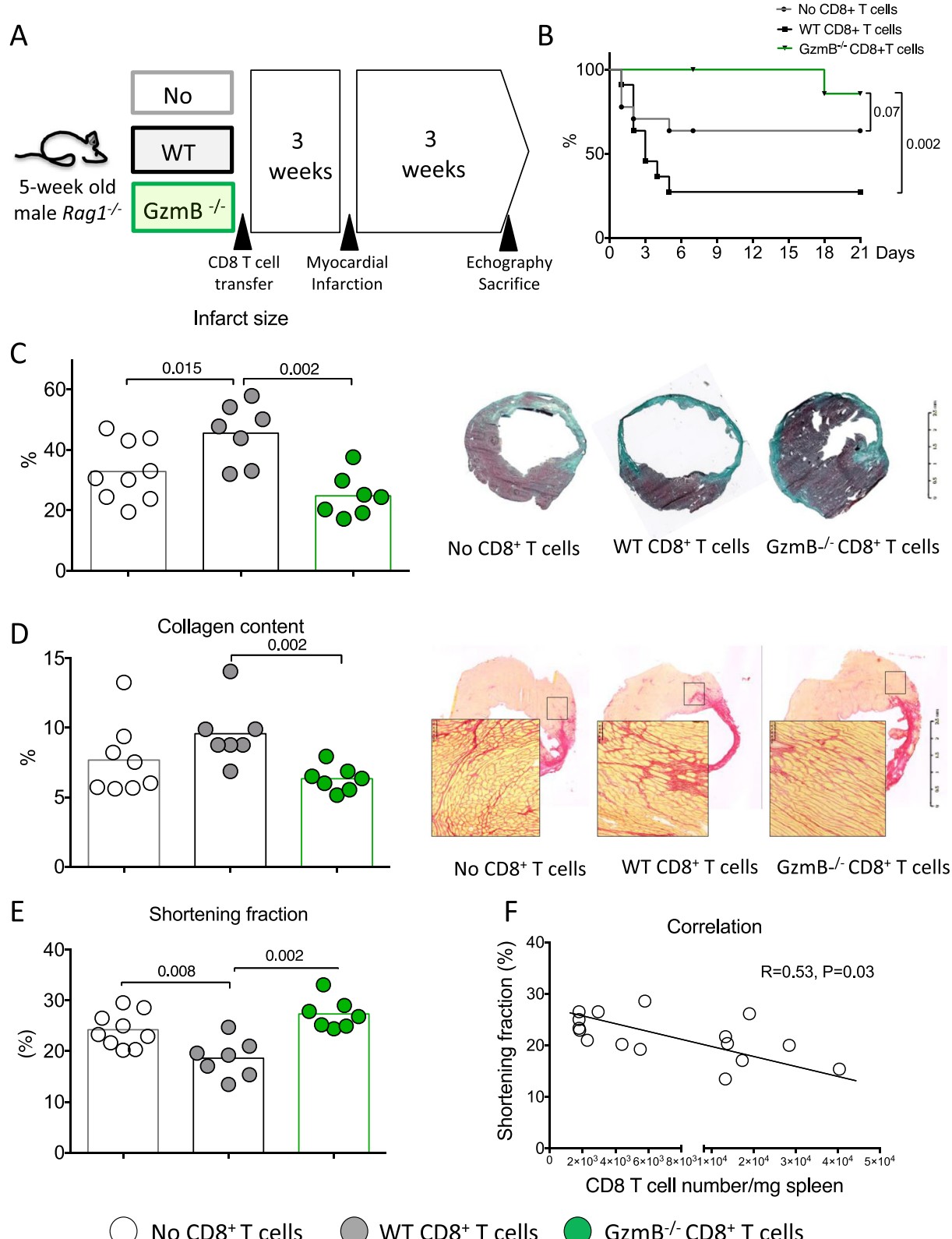

obtained as early as 6 h following mAb injection in mice (Supplementary Fig. 8). Based on this observation, we designed a protocol (Supplementary Fig. 35) including 1 control group and 2 CD8-depleted groups receiving anti-CD8 antibody at 2 different time points to obtain either High depletion (>95% depletion at day 1 after MI) or low depletion (60% depletion at day 1 after MI)

(Fig. 6A, B). As previously validated by our group[25], coronary occlusion was maintained for 40 min (Supplementary Fig. 35) leading to transmural myocardial infarct. At day 14, we observed no difference in the infarct size between low CD8 depletion and control groups but we found a significant reduction of infarct size in high-CD8-depletion group ($-60\%$ vs control, $P < 0.01$)

**Fig. 5 CD8$^+$ T lymphocytes trigger adverse ventricular remodeling and alter heart function through the production of Granzyme B. A** *Rag1$^{-/-}$* mice injected with either CD8-depleted splenocytes (White) or CD8 cell-depleted splenocytes re-supplemented with WT (gray) or *GzmB$^{-/-}$* CD8$^+$ T cells (green), 3 weeks before MI. **B** Survival curves following MI (from 3 experiments, No CD8 $n = 15$, WT CD8 $n = 18$, and *GzmB$^{-/-}$* CD8 $n = 8$). **C** Representative photomicrographs and quantitative analysis of infarct size. **D** Collagen content in the peri-infarct area in the 3 groups of mice, scale bar 100 μm. Results are pooled from three independent experiments including surviving mice (no CD8 $n = 9$, WT CD8 $n = 7$, and *GzmB$^{-/-}$* CD8 $n = 7$). **E** Echocardiography analysis after 21 days of MI and assessment of LV shortening fraction (SF) in the 3 groups of mice (no CD8 $n = 9$, WT CD8 $n = 7$, and *GzmB$^{-/-}$* CD8 $n = 7$). **F** Correlation between CD8$^+$ T cell number in the spleen at day 21 and LV shortening fraction. Data from CD8-depleted splenocytes or CD8 cell-depleted splenocytes re-supplemented with WT CD8$^+$ T cells have been included ($n = 16$). *P* values were calculated using two-tailed Kruskal-Wallis test (**C**, **D**, **E**). Difference in survival was evaluated using log-rank test (**B**) and correlation was studied using Spearman's test (**F**).

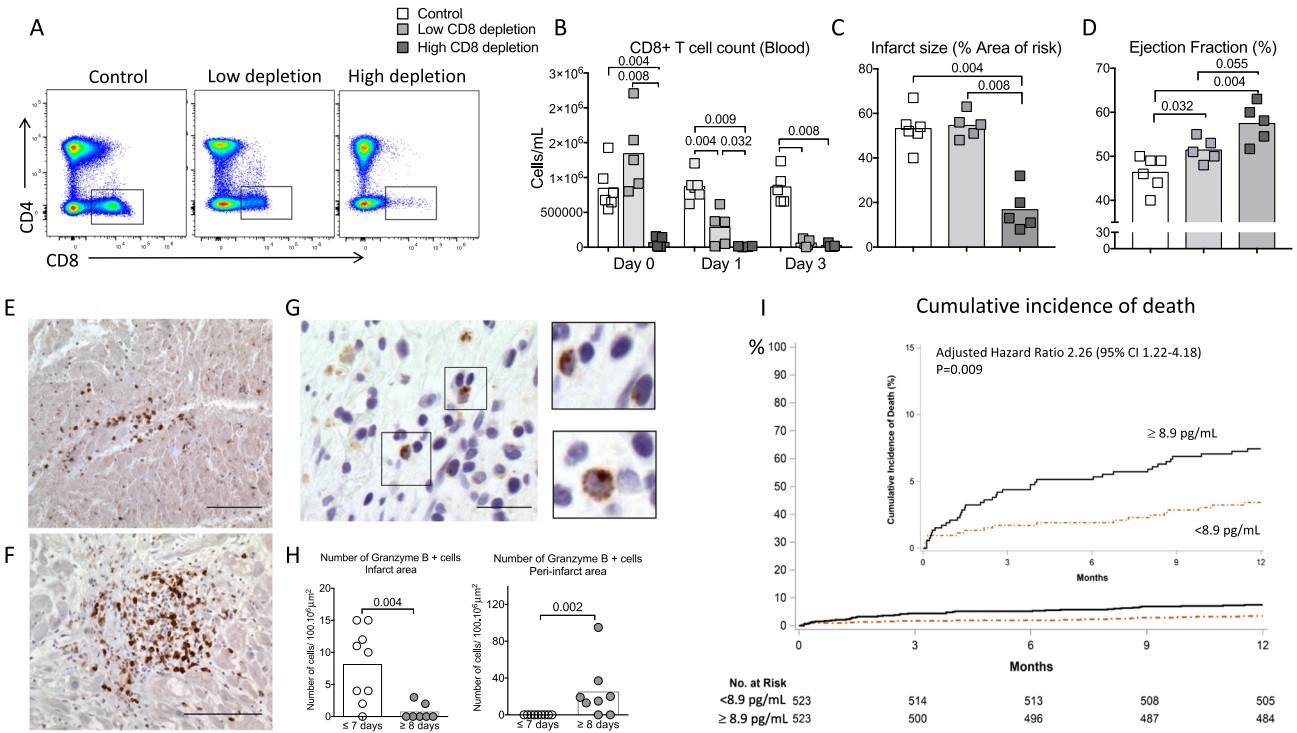

**Fig. 6 Pathogenic role of CD8$^+$ T cells in a model of myocardial ischemia/reperfusion in pig and relevance to the human disease. A** Flow cytometry analysis of blood CD4$^+$ T and CD8$^+$ T subsets at day 1 after MI in control (PBS, CTR, $n = 6$), low (light gray, $n = 5$), and high (dark gray, $n = 5$) CD8-depleted groups. **B** Quantification of CD8$^+$ T cell count in the blood at baseline, day 0, day 1, and day 3 after MI. **C** Representative picture and quantification of infarct size at day 14 in control, low, and high CD8-depleted groups. **D** Quantitative evaluation of left ventricle ejection fraction (Simpson) of CTR or CD8-depleted pigs. **E**, **F** Detection of CD8$^+$ T cells (brown) in human heart biopsy of MI patients, using immunohistochemistry at day 3 (**E**, upper) and day 8 (**F**, lower) after MI, scale bar 50 μm. **G** Detection of GRANZYME B+ cells (brown) in human heart biopsy of MI patients, using immunohistochemistry, scale bar 20 μm. **H** Quantification of GRANZYME B+ cells (brown) in human heart biopsy of MI patients, using immunohistochemistry at different time points ($n = 9$ day ≤ 7, $n = 8$ day ≥ 8). **I** Survival according to baseline circulating GRANZYME B level (< or > median value) in patients with acute MI ($n = 1046$). High level of GRANZYME B at the admission for acute MI were independently predictive of death after 1 year of follow-up after multiple adjustments (see Methods and Supplementary Table 3). HR = hazard ratio. *P* values were calculated using two-tailed Mann-Whitney test (**H**) or Kruskal-Wallis test (**B–D**).

(Fig. 6C). Finally, both high and low CD8 depletion significantly improved the LV systolic function but the beneficial impact of CD8 depletion was more important in high-depletion group (Fig. 6D and Supplementary Figs. 36 and 37). In addition, CD8 depletion led to decreased Granzyme B plasma levels (Supplementary Fig. 38). Taken together, these results confirmed the pathogenic role of CD8$^+$ T cells in reperfused acute MI in a large animal model.

**Granzyme B and CD8$^+$ T cells in human MI.** In human heart biopsies obtained from acute MI patients (Supplementary Table 1), we detected CD8$^+$ T cell infiltration in the ischemic heart tissue at day 3 (Fig. 6E) and day 8 after MI (Fig. 6F). GRANZYME B+ cells were mainly detected in the infarct area

within the first week of MI, but predominated in the peri-infarct region after day 7 (Fig. 6G, H).

Finally, we addressed the relevance of these findings to the human disease by assessing the relationship between circulating GRANZYME B levels and clinical outcomes among those 1046 patients (Supplementary Table 2) who contributed to a serum bank in FAST-MI, a nationwide cohort of consecutive adults with ST-segment-elevation or non-ST-segment-elevation MI hospitalized at intensive care unit with symptom onset ≤48 h, in 213 centers representing 76% of French centers managing acute MI patients (NCT01237418). Interestingly, we found that acute MI patients with high circulating levels of GRANZYME B (>median 8.9 pg/mL) at their admission were at higher risk of death after 1 year of follow-up compared to patients with low levels even after adjustment for several multivariable risk factors (Supplementary

Table 3) (adjusted hazard ratio, HR = 2.26, 95% CI = 1.22–4.18, $P = 0.009$) (Fig. 6I).

## Discussion

CD8[+] T cells play a critical role in anti-viral[20] and anti-tumor[19] immune responses. Recently, studies identified novel roles for CD8[+] T lymphocytes in sterile chronic low-grade inflammation such as atherosclerosis[26]. However, the contribution of CD8[+] T cells to the inflammatory response secondary to other forms of acute injury, particularly post-ischemic injury is still poorly defined. Our results unravel a critical role for CD8[+] T cell-dependent Granzyme B production in the pathogenic response to acute MI injury.

In a mouse model of MI with permanent coronary ligation, we showed that activated CD8[+] T cells expressing CD69 and CD107a markers were recruited within the ischemic myocardium. Such CD8[+] T cell infiltration in the heart has also been observed in experimental transient ischemia[27]. In humans, CD8[+] T cells were identified in the ischemic heart tissue at an early stage after MI, but predominated in the peri-infarct region 1 week after MI. We also observed that CD8[+] T cell recruitment in cardiac muscle was partially mediated by CD4[+] T lymphocytes. Similarly, in the context of myocarditis, CD4[+] T cells have been shown to drive CD8[+] T cell trafficking through the release of IL-21[16].

Combining "loss of function" using CD8 cell-depleting treatment with anti-CD8 mAb and "gain of function" strategy based on CD8[+] T cell supplementation in $Rag1^{-/-}$ mice, we showed that CD8[+] T cells promoted deleterious post-ischemic cardiac remodeling at early (day 21) and late stages (day 56) in a murine model of permanent coronary occlusion, as well as in a pig model of coronary ischemia-reperfusion. In mice, CD8[+] T cells fostered adverse ventricular remodeling through their pro-apoptotic effects. On the same note, CD8[+] T cells can trigger neuron cell death in vitro and CD8[+] T cell depletion limited brain apoptotic area in a mouse model of stroke[28]. Effector mechanisms likely involved in CD8[+] T cell-mediated cell death include Perforin/ Granzyme pathway, two death receptor molecules of the TNF receptor family, such as Fas and pro-inflammatory cytokines, including Interferon-γ. We focused here on the role of Granzyme B, a serine protease released by cytotoxic T cells activating apoptosis pathways and triggering caspase activation indirectly. This protease was detected in the peri-infarct area at early time points following coronary occlusion and co-localized with both infiltrating CD8[+] T cells and TUNEL+ cells. Interestingly, we found that Granzyme B was detected in T cell cytoplasm in non-ischemic area, whereas Granzyme B was found around CD8[+] T cells in the peri-infarct area, suggesting a mechanism of local degranulation. Granzyme B has been previously identified as a major toxic protein in auto-immune diseases such as diabetes[29], as well as in cardiovascular diseases such as stroke[28]. In our study, Granzyme B mediated the deleterious effect of CD8[+] T cells after MI. In vitro, we showed that activated purified CD8[+] T cell induced cardiomyocyte apoptosis, and cell death was abolished when CD8[+] T cells where isolated from $GzmB^{-/-}$ mice. Furthermore, the deleterious cardiac effects of CD8[+] T cell reconstitution in $Rag1^{-/-}$ mice were blunted when animals were repopulated with $GzmB^{-/-}$ CD8[+] T cells. Importantly, GRANZYME B-expressing T cells were detected in human heart tissue of MI patients, and we found that high plasma levels of Granzyme B within 48 h of admission for acute MI was associated with increased risk of 1-year mortality. On the same note, the number of circulating CD8[+] CD28[+] T cell was associated with more severe ischemic heart dysfunction in a small cohort of MI patients[30]. Hence, one can speculate that GRANZYME B plasma levels could be used as a biomarker to select patients who may

benefit from anti-CD8-depleting antibody in acute MI. Such personalized approach should be tested in the future when human anti-CD8 mAb treatment will be available.

CD8[+] T cell depletion led to a switch of the immune response within the ischemic heart toward a less inflammatory profile. Local reduction of pro-inflammatory cytokines was unlikely due to decreased leukocyte recruitment, since NK, NKT, neutrophil, classical monocyte, B cell as well as CD4[+] T cell counts were similar in the infarcted heart of control and CD8-depleted groups. Local deviation of the immune response following CD8 treatment was probably due to a switch of the macrophage profile toward an anti-inflammatory phenotype. Inflammatory macrophages in a classic activation mode, dominated at days 1–3 post-MI, whereas reparative macrophages in an alternative activation mode were the major subsets at days 5–7 post-MI in mouse heart. Pro-inflammatory macrophages secrete cytokines, chemokines, growth, and matrix metalloproteinase (MMPs), whereas anti-inflammatory macrophages are pro-reparative[31]. Local reduction of cardiomyocyte apoptosis and consecutive DAMPs release following CD8 depletion may explain such deviation of macrophage phenotype toward an anti-inflammatory profile[32]. Of note, in Cd8[atm1mak] mice, a genetically modified mouse model characterized by CD8 T cell deficiency, CD8[+] T cells have been shown to play a dual and contradictory role. Indeed, animals lacking functional CD8[+] T cells had increased cardiac rupture despite having better overall survival after MI[33]. These results should be analyzed with caution since an effect of Cd8a gene mutation on other immune cells, as well as confusing genetic compensation, could not be ruled out in this mouse model[34]. In this line of reasoning, an increased Th1 responses and an impaired B cell humoral activity have already been reported in Cd8[atm1mak] mice[35]. In our study, conclusions are supported by different complementary approaches including a very specific monoclonal-targeted strategy with selective CD8[+] T cell depletion in immunocompetent adult mice.

Decreased apoptosis and generation of secondary necrotic cells most likely accounted for the low inflammatory profile observed in infarcted hearts of CD8-depleted mice. Similarly, $GzmB^{-/-}$ mice exhibited reduced apoptosis and pro-inflammatory cytokine expression. Finally, we showed that TCR engagement was critical for CD8[+] T cell cytotoxic activity. CD8 depletion in OT-I mice had no effect on post-ischemic cardiac remodeling. Repopulation of $Rag1^{-/-}$ mice with purified OT-I CD8[+] T cells had less deleterious consequences than those of WT CD8[+] T cells, in terms of infarct size and LV systolic function. In line with these findings, transfer of OT-I CD8[+] T cells in $CMy$-$mOva$ mice expressing membrane-bound ovalbumin specifically on cardiomyocytes led to elevated Granzyme B content in the ischemic heart, high mortality rate, and pronounced deleterious cardiac remodeling in survivors after MI. Hence, our results indicate that, in acute MI, CD8[+] T cells are activated by autoantigens presented by MHC class I molecules. Further studies are required to identify the specific peptides recognized by CD8[+] T cells. Cardiac contractile proteins may represent such candidates. Previous works have shown that α-myosin heavy chain is a pathogenic auto-antigen for CD4[+] T cells in a mouse model of spontaneous myocarditis[36].

Our study delineates an important contributory role for CD8[+] T cells during the immune-inflammatory responses that lead to tissue damage following acute MI. Conversely to B cells or CD4[+] T cells, CD8[+] T cells have no major role in regulating immune cell recruitment within the site of injury. CD8[+] T cells are quickly recruited within the ischemic heart tissue and are activated in an antigen-specific manner. Activated CD8[+] T cells release Granzyme B that has cytotoxic activity in the heart promoting cardiomyocyte death and a sterile pro-inflammatory immune

response. As a consequence, CD8$^+$ T cell depletion substantially limits heart cell apoptosis, myocardial inflammation, infarct size, and finally improves myocardial function. The positive association of GRANZYME B circulating levels with poor outcome in patients with acute MI strongly supports the clinical relevance of our findings to the human disease.

Experiments in female pigs were performed to confirm the pathogenic role of CD8$^+$ T cells in a large model of heart ischemia-reperfusion, mimicking percutaneous transluminal coronary angioplasty during human acute MI. We found beneficial effect of CD8$^+$ depletion in this pig model but the mechanism of protection was probably different between high- and low-CD8-depletion groups. High CD8 depletion may be protective through a reduction of cardiomyocyte apoptosis and infarct size, a mechanism that we previously found in mouse models of permanent coronary artery occlusion. Beneficial effect of low CD8 was unlikely due to decreased cardiac cell apoptosis but may be due to a protective effect on cardiomyocyte function. Such a hypothesis was supported by in vitro experiments showing that low number of activated CD8$^+$ T cells are able to impair cardiomyocyte contractility in a Granzyme B-dependent manner without inducing cell death. Balloon occluder implantation to induce transient coronary artery occlusion in pig was done surgically 14 days before heart ischemia and long-term surgery-induced inflammation could not be ruled out.

We believe that the present study paves the way for the development of therapeutic approaches based on the administration of humanized CD8-depleting antibodies during the acute phase of MI. Because of pharmacodynamic characteristics of anti-swine monoclonal anti-CD8 antibody, anti-CD8 mAb has to be injected before ischemia-reperfusion procedure to induce high CD8 depletion. Obviously, such a therapeutic strategy could not be used in human MI to limit deleterious post-ischemic cardiac remodeling, but underscores the need for developing an anti-human anti-CD8 mAb that induces full and rapid CD8$^+$ T cell depletion to translate cardiac benefit from animal to human.

## Methods

**Mice models**. All experiments were conducted according to the French veterinary guidelines and those formulated by the European Community for experimental animal use and were approved by the Institut National de la Santé et de la Recherche Médicale. The protocol was approved by the ethical committee CEEA34 Université de Paris (APAFIS #10554-2017041016471398). All the animals were maintained under identical standard conditions (housing, regular care and normal chow). Mice were maintained in isolated ventilated cages under specific pathogen-free conditions.

**Myocardial infarction**. All mice were on full C57Bl/6J background. C57BL/6 (Janvier, France), *GzmB*$^{-/-}$, *Rag1*$^{-/-}$, and *OT-I* mice (Jackson, USA) as well as *CMy-mOva* mice (A. Lichtman's lab, USA)[18] were 9-week-old at the time of MI. Myocardial infarction was induced by left anterior descending coronary artery ligation[37]. Mice were anesthetized using ketamine (100 mg/kg) and xylazine (10 mg/kg) via intraperitoneal (i.p.) injection, then intubated and ventilated using a small animal respirator. The chest wall was shaved and a thoracotomy was performed in the fourth left intercostal space. The LV was visualized, the pericardial sac was then removed, and the left anterior descending artery was permanently ligated using a 7/0 non-absorbable monofilament suture (Peters surgical, France) at the site of its emergence under the left atrium. Significant color changes at the ischemic area were considered indicative of successful coronary occlusion. The thoracotomy was closed with 6/0 non-absorbable monofilament sutures (Peters surgical, France). The same procedure was performed for sham-operated control animals except that the ligature was left untied. The endotracheal tube was removed once spontaneous respiration resumed, and animals were placed on a warm pad maintained at 37 °C until the mice were completely awake.

To determine the recruitment and activation of CD8 T+ lymphocytes in the ischemic heart muscle after MI, *C57bl6* WT mice were subjected to permanent coronary ligation; sham *C57bl6* WT mice were used as control in this set of experiments (the same MI procedure was performed for the sham-operated mice except that the coronary ligation was not tied). Mice were sacrificed at different time points for flow cytometry and histological analysis. In order to assess the role of CD4$^+$ T cells in the recruitment of CD8$^+$ T cells in ischemic cardiac muscle,

*C57bl6* WT mice were injected i.p. with a depleting rat anti-mouse CD4 monoclonal antibody (100 μg/mouse, Biotem Clone YTS 169.4)[38] or with a rat IgG2b isotype control (R&D Systems; Abingdon, UK) 24 h before MI. These mice were sacrificed 1 day or 3 days after the surgery for flow cytometry analysis.

To evaluate the role of CD8$^+$ T cells in post-ischemic cardiac remodeling, *C57bl6* WT mice were injected i.p. with a depleting rat anti-mouse CD8 monoclonal antibody (100 μg/mouse, Biotem Clone YTS 169.4)[38] or with a rat IgG2b isotype control (R&D Systems; Abingdon, UK) 1 h after coronary artery ligation. In order to maintain cell depletion for an extended period, mice were re-injected at day 7 and day 14 after MI. Animals were sacrificed 24 h, 3 days, 7 days, and 14 days after surgery for flow cytometry and histological analysis.

To investigate if TCR engagement aggravates CD8 cardiac cytotoxicity, we used two genetically modified mice models: (1) C57BL/6-Tg(TcraTcrb)1100Mjb/J mouse is also known as OT-I (Charles River). The transgenic T cell receptor of OT-1 mouse was designed to recognize ovalbumin peptide residues in the context of H2Kb (CD8 co-receptor interaction with MHC class I). This results in MHC class I-restricted, ovalbumin-specific. The CD8 T cells of OT-I mouse primarily recognize OVA peptide when presented by the MHC I molecule; (2) CMy-mOva mouse is a transgenic mouse line with cardiomyocyte restricted membrane-bound ovalbumin expression[18]. OT-I mice were injected i.p. with a rat anti-mouse CD8 monoclonal antibody (100 μg/mouse, Biotem Clone YTS 169.4)[38] or with a rat IgG2b isotype control (R&D Systems; Abingdon, UK) 1 h after MI. In order to maintain cell depletion for an extended period, mice were re-injected at day 7 and day 14 after MI. Next, 6-week-old immunodeficient *Rag1*$^{-/-}$ mice were used as recipients for CD8$^+$ T cell repopulation studies 21 days prior MI. Purified WT CD8$^+$ T cells or OT-I CD8$^+$ T cells were obtained from *C57bl6* WT mice or OT-I mice, respectively. Cells were resuspended in phosphate-buffered saline (6 × 10$^6$ cells/200 μL/ mouse) and transferred to *Rag1*$^{-/-}$ mice by i.v. injection. Finally, 8 weeks old *CMy-mOva* mice received either 6 × 10$^6$ WT CD8$^+$ T lymphocytes or 6 × 10$^6$ OT-I CD8$^+$ T lymphocytes 3 days before MI. In all these experiments, OT-I, *Rag1*$^{-/-}$, and *CMy-mOva* mice were sacrificed 21 days after MI for histological analysis.

In order to show that Granzyme B specifically released by CD8$^+$ T cells mediates cardiac cytotoxicity, immunodeficient *Rag1*$^{-/-}$ mice received either 4 × 10$^6$ CD8$^+$ T cell-depleted splenocytes (Including all spleen leukocyte subsets except CD8$^+$ T cells), CD8$^+$ T cell-depleted splenocytes re-supplemented with 6 × 10$^6$ WT purified CD8$^+$ T lymphocytes, or 6 × 10$^6$ purified *GzmB*$^{-/-}$ CD8$^+$ T lymphocytes. Mice were then allowed to recover for 3 weeks before they were challenged with MI. Repopulation was checked at day 3 following MI in the spleen and the heart. Recipient *Rag1*$^{-/-}$ mice were sacrificed 21 days after surgery for histological analysis.

**Echocardiographic measurements**. Transthoracic echocardiography was performed at 9, 21, and 28 days after surgery using an echocardiograph (ACUSON S3000™ ultrasound, Siemens AG, Erlangen Germany) equipped with a 14-MHz linear transducer (1415SP). The investigator was blinded to group assignment. Animals were anesthetized by isoflurane inhalation. Two-dimensional parasternal long-axis views of the LV were obtained for guided M-mode measurements of the LV internal diameter at end diastole (LVDD) and end systole (LVDS), as well as the interventricular septal wall thickness, and posterior wall thickness at the same points. Fractional shortening percentage (%FS) was calculated by the following formula: %FS = [(LVDD − LVDS)/LVDD] × 100. Invasive left ventricular hemodynamic indices were measured in anesthetized mice with a 2F dual-field combination pressure-conductance catheter (model SPR-819, Millar Instruments) as previously described[39].

**CD8$^+$ T cell purification and transplantation**. CD8$^+$ T cells were isolated from C57BL/6J, *GzmB*$^{-/-}$, or OT-I spleens and purified using a CD8$^+$ T cell isolation kit (Miltenyi Biotec, Paris, France) according to the manufacturer's protocol. In brief, CD8$^+$ T cells were negatively selected using a cocktail of antibody-coated magnetic beads (CD4, CD11b, CD11c, CD19, CD45R (B220), CD49b (DX5), CD105, Anti-MHC-class II, Ter-119, and TCRγ/δ) followed by cell separation using LS magnetic columns (Miltenyi Biotec; Paris, France), yielding CD8$^+$ T cells with >95% purity. Then cells were: (a) intravenously injected 21 days prior MI in *Rag1*$^{-/-}$ mice or 3 days before MI in *CMy-mOva* mice and (b) cultured for in vitro experiments.

**CD8$^+$ T cell activation**. CD8$^+$ T cells were isolated as described above and cultured in activation medium containing RPMI-1640 supplemented with 10% fetal bovine serum, 100 U/mL penicillin, 100 U/mL streptomycin (Gibco, Thermo-Fisher), 50 U/mL IL2, 10 ng/mL IL12 (R&D Systems; Abingdon, UK), and dynabeads mouse T activator CD3/CD28 (Gibco, ThermoFisher) in a ratio of 1:1 with respect to CD8$^+$ T cell count.

**HL1 cardiomyocytes culture**. Cells were seeded into 24-well plates (coated with gelatin-fibronectin (Sigma-Aldrich) overnight, 37 °C) at a density of 20 × 10$^3$ cells/mL/well and maintained for 2 days in Growth Medium containing Claycomb Medium (Sigma-Aldrich) supplemented with 10% fetal bovine serum, 100 U/mL penicillin, 100 U/mL streptomycin (Gibco, ThermoFisher), 1% norepinephrine 10 mM (Sigma-Aldrich), and 1% L-Glutamine 100X (Sigma-Aldrich).

**Co-culture and apoptosis experiment.** Activated CD8$^+$ T cells were co-cultured with HL1 cardiomyocytes in different ratios (1:1, 1:5, and 1:10) with respect to cardiomyocyte count. Following co-culture for 24 h, CD8$^+$ T cells were removed and HL1 cardiomyocytes were labeled with the addition of 5 μM Caspase-3/7 Green Apoptosis Reagent (Incucyte, Welwyn Garden, UK) to the growth medium. After 30 min at 37 °C in dark, apoptosis in real-time was evaluated with a microscope Nikon Eclipse, Ti. Time-lapse images were captured at 10 min intervals.

**Co-culture and cardiomyocyte contractility.** Cardiomyocytes were isolated using a simplified Langendorff-free method as previously reported[40] with minor modifications. Briefly, C57/BL6J mice aged 10–12 weeks were anesthetized, their chests were washed with 70% ethanol and opened to expose the heart. The descending aorta was cut and 7 mL of EDTA buffer (130 mmol/L of NaCl; 5 mmol/L of KCl; 0,5 mmol/L of Monosodium Phosphate; 10 mmol/L of HEPES, 10 mmol/L of Glucose, 10 mmol/L of 2,3-Butanedione 2-monoxime, 10 mmol/L of Taurine, 5 mmol/L of EDTA) was injected into the right ventricle. Ascending aorta was then clamped using atraumatic forceps and the heart was transferred to a 60-mm dish containing fresh EDTA buffer. The petri dish was fixed on a heating block to reach a temperature of 37 °C. The heart was then perfused in the LV with a set up using a 26 G needle connected to a line and a syringe and then subjected to sequential injections of 10 mL of EDTA buffer, 10 mL of perfusion buffer (130 mmol/L of NaCl; 5 mmol/L of KCl; 0,5 mmol/L of Monosodium Phosphate; 10 mmol/L of HEPES, 10 mmol/L of Glucose, 10 mmol/L of 2,3-Butanedione 2-monoxime, 10 mmol/L of Taurine, 1 mmol/L of MgCl$_2$), and 45 mL of collagenase buffer (105 U/mL of Collagenase II; 130 U/mL of Collagenase IV; 10 U/mL of Protease XIV; diluted in perfusion buffer) using a syringe pump fixed at a steady rate of 80 mL/h. At the end of the digestion, the atria and remaining aorta were excised and discarded. Cellular dissociation was completed by gentle mechanical trituration, and enzyme activity was stopped by adding 5 mL of stop buffer (10% FBS; 100 mmol/L of CaCl$_2$; diluted in perfusion buffer). Cell suspension was passed through a 100-μm filter to remove non-digested tissue. Calcium concentration in the cardiomyocytes was gradually restored to physiological levels by adding an increased quantity of calcium buffer 5 times every 4 min. Concentration of viable rod-shaped cardiomyocytes was quantified using a hemocytometer. The cardiomyocytes were then resuspended in prewarmed plating media (M199, Gibco; 5% FBS; 10 mmol/L of 2,3-Butanedione 2-monoxime, Sigma Aldrich) and plated at a density of 20,000 cardiomyocytes/well, onto laminin (50 μg/mL) precoated 25 mm × 25 mm glass coverslips in a humidified tissue culture incubator (37 °C, 5% CO$_2$). After 2 h, media was changed to fresh, prewarmed culture media (M199, Gibco; 5% BSA; 1X ITS supplement, Sigma Aldrich; 10 mmol/L of 2,3-Butanedione 2-monoxime, Sigma Aldrich. 1X CD lipid. Chemically Defined Lipid Concentrate, Thermo Scientific). Activated WT or GzmB$^{-/-}$ purified spleen CD8$^+$ T lymphocytes were co-cultured with cardiomyocytes in 1:3 ratio with respect to cardiomyocyte count. Following co-culture for 12 h, CD8$^+$ T cells were discarded and the medium replaced with prewarmed culture media. Sarcomere shortening was measured on myocytes incubated in fresh prewarmed culture media and field stimulated at 1 Hz (20 V, 1 ms). Data were recorded at room temperature (20 °C) using Ionoptix Imaging System coupled to a microscope (×40; ZEISS) and analyzed using SarcLen software (IonWizard 6 Software; Ionoptix, Milton, MA, USA).

**Histopathological and immunofluorescence analyses.** Cardiac healing following MI was assessed at different time points. Hearts were excised, rinsed in PBS, and frozen in liquid nitrogen. For tissue morphology analysis after MI, hearts were cut along their length into 7-μm thick cardiac muscle cryosections (CM 3050S, Leica). Serial sections were mounted on microscope slides ($n = 10$), each section being spaced from the next one by 550 μm. In this way, 6–7 cuts/heart (site where the suture was observed), allowed entire heart tissue analysis. Masson's trichrome and Sirius Red stainings were performed for infarct size and interstitial fibrosis evaluation, respectively. Infarct size was calculated as a percentage of infarct area to total LV circumference. The collagen volume fraction was calculated as the ratio of the total area of interstitial fibrosis to the myocyte area in the entire visual field of the section.

Heart sections for immunofluorescence analysis were fixed with paraformaldehyde 4%, permeabilized using 0.2% Triton X100 in Phosphate Buffer Solution (PBS) 30 min at room temperature, blocked with PBS-T (0.2% Triton X100, 10% goat serum, 0.2% BSA in PBS) for 1 h, and incubated with primary antibodies diluted in PBS-T overnight at 4 °C: CD8 T lymphocyte infiltration was detected using an anti-CD8 antibody (Abcam, 1:200); Granzyme B detection in ischemic heart tissue was performed using an anti-Granzyme B antibody (R&D Systems 1:100); Sections were washed with PBS and incubated with a mixture of appropriate secondary antibodies for 1 h at room temperature: Cyanine 3 Goat anti-rat and Cyanine 5 Donkey anti-goat, respectively (Jackson Immunoresearch). 4′,6-diamidino-2-phenylindole (DAPI) was used to counterstain cell nuclei. To evaluate cell apoptosis (day 3 post-MI), immunofluorescence analyses were performed using a TUNEL assay kit (Roche Diagnostics, Meylan, France) according to manufacturer's instructions. The digital images of immunofluorescence were acquired with a Zeiss Axioimager Z2 Apotome. Two microscopic fields in the infarcted area (in the case of CD8$^+$ T cells and Granzyme B staining) and in both sides of peri-infarct area (in the case of TUNEL staining) from each heart were examined using ImageJ64. All illustrative stainings were repeated once (totally two independent experiments, $n = 5$/group/experiment).

**TTC staining.** To evaluate infarct size at early time points (3 days after MI), 2,3,5-triphenyltetrazolium chloride (TTC) was used. TTC staining identifies metabolically active tissue and is helpful to measure tissue viability. Hearts were removed and sectioned into 2 mm thick slices in a semi frozen state. The slices were then immersed for 40 min in 1% TTC solution at 37 °C. Once the color had been established, slices were fixed in 4% PAF overnight at room temperature. Slices were photographed and infarcted heart tissue areas were calculated using ImageJ64 blindly.

**Ex vivo reflectance epifluorescence imaging.** Mice were anaesthetized with isoflurane and received intravenously 150 μL of a fluorescent imaging probe MMP-sense 680 (NEV 10126, PerkinElmer) 24 h before sacrifice. This agent is optically silent in its un-activated state and becomes highly fluorescent following activation by MMPs including MMP-2, −3, −9, and −13. Images of ischemic hearts were acquired using a fluorescence molecular imaging system (FMT 2500TM, VisEn Medical)[41].

**Cell suspension preparation for flow cytometry.** Mice were sacrificed at 12 h and on days 1, 3, 5, 7, 14, and 21 post-MI ($n = 5–10$ mice per time point). Peripheral blood was drawn via inferior vena cava puncture using a syringe primed with heparin solution. For blood staining, erythrocytes were lysed using BD FACS lysis solution (BD Biosciences). Spleens were surgically removed and dissociated, obtaining a single cell suspension that was filtered through 40-μm nylon mesh (BD Biosciences). Cell suspensions were centrifuged at 400$g$ for 15 min at 4 °C. Red blood cells were removed using a red blood lysis buffer (Sigma-Aldrich) and splenocytes were washed with PBS supplemented with 3% FBS. LV of infarcted and control mice were harvested, minced with fine scissors, and gently passed through a Bel-Art Scienceware 12-Well Tissue Disaggregator (Fisher Scientific). Cells were collected, filtered through 40-μm nylon mesh (BD Biosciences), and centrifuged at 400$g$ for 15 min. Red blood cells were removed by incubation in red blood cell lysis buffer (Sigma-Aldrich) and single cell suspension were washed with PBS supplemented with 1% fetal bovine serum.

**Flow cytometry.** General characteristics of antibodies are summarized in Supplementary Tables 4 and 5. Surface stainings were performed before permeabilization when intracellular labeling were necessary. Cells were incubated in Fixation/Permeabilization buffer (eBiosciences) for 45 min, washed with permeabilization buffer (eBiosciences), and stained with the appropriate marker for intracellular stainings. Forward scatter (FSC) and side scatter (SSC) parameters were used to gate live cells excluding red blood cells, debris, and cell aggregates in total splenocytes. Cells were analyzed using a BD CantoII or BD LSRII flow cytometer (BD Biosciences).

**Quantitative real-time PCR.** Quantitative real-time PCR was performed on a Step-one Plus (Applied Biosystems) qPCR machine. GAPDH was used to normalize gene expression. The primer sequences are described in Supplementary Table 6.

**Myocardial infarction in pigs.** The studies were conducting according to relevant ethical guidelines and the protocol was approved by the ethical committee ComEth Anses/EnvA/UPEC (APAFIS #3841-2016012815406796). Pigs (Land race White crossed, Lebeau, Gamblais France) were pretreated with azaperon (8 mg/kg IM) and atropine (50 μg/kg). After administration of ketamine (20 mg/kg i.v.), anesthesia was induced and maintained by isoflurane (2.5%). Animals also received morphine (2 mg/kg) intravenously. After administration of rocuronium (2,5 mg/kg), fluid-filled Tygon catheters was implanted in the descending thoracic aorta, left atrium, and pulmonary artery for the measurement of arterial blood pressures, administration of lidocaine in case of arrhythmias, and drug administration, respectively. A balloon occluder was inserted around the left descending coronary artery to induce its occlusion and reperfusion. Catheters were exteriorized between the scapulae. Chest was closed in layers. For postoperative care, animals were treated with fentanyl (patch, 100 μg/h for 2 days) and long-acting amoxicillin (15 mg/kg every 2 days for 10 days). Two weeks after surgery, animals were sedated with tiletamine and zolazepam (both 10 mg/kg) and then anesthetized with thiopental (50 mg/kg/h for 25 min). The animals were intubated and properly ventilated and the balloon occluder was inflated for 40 min to occlude the left descending coronary artery followed by reperfusion. PBS or depleting anti-CD8 monoclonal antibody (76-2-11, IgG2a mouse anti-swine, 15 mg/kg)[24] was administered intravenously either 3 days before coronary occlusion (high CD8 depletion) or 60 min after reperfusion (low CD8 depletion) (Supplementary Fig. 32). The animals returned to their cage for 14 days. At day 14 after MI, animals underwent echocardiographic examination. This was performed and analyzed by the same sonographer blinded to treatment information. Using a Vivid 7 ultrasound unit (General Electric Medical System, Horten, Norway) under ECG monitoring, echocardiographic images were acquired in pigs placed in standing position in a sling and sedated with propofol (induction dose: 2 mg/kg, i.v.; maintenance dose: 5 mg/kg/h), using a 4.3-MHz transducer. Echocardiographic images at apical 4-chamber and 2-chamber vies were acquired blindly. For each image, three consecutive cardiac cycles were stored digitally for offline analysis (EchoPac 6.0, GE Healthcare). Left ventricular (LV) ejection fraction

was calculated by the biplane Simpson method using both the apical 4-chamber and 2-chamber views. At sacrifice, pigs were re-anesthetized and the balloon occluder was inflated again. Crystal violet solution (0.8%) was administered intravenously to delineate the previously occluded vascular bed (area at risk, AAR). The heart was excised, and the LV was cut into slices that were weighed and incubated in 1% TTC, 37 °C) to detect MI. Slices were overnight fixed in 5% formaldehyde and then photographed. Using a computerized planimetric program (ImageJ, NIH, Bethesda, MD, USA), the AAR, the infarcted and salvaged zones were delimitated and quantitated blindly. The AAR was identified as the non-violet region and was expressed as a percentage of the LV weight. Infarcted area was identified as the TTC-negative zone and was expressed either as a percentage of AAR. The viable tissue or salvaged area was expressed as percentage of AAR.

**Immunostaining on human ischemic heart tissue**. All the patients gave informed consents for the surgical procedure. All the patients had LV assist device (Heart-Mate II or HeartWare) implantation for severe cardiac failure. For implantation of the device, the apex of the LV was opened and tissue samples were excised. Tissues were fixed in formalin and paraffin embedded. The tissue sections were stained with H&E for pathological diagnostic procedure allowing dating of MI in addition to clinical dating. On remnant sections, immunohistochemistry was performed with anti-CD8 (clone SP239, Abcam) diluted at 1:500 and anti-GRANZYME B (clone GrB-7, Dako) diluted at 1:150 using an automated system (Roche Diagnostics France, Meylan, France). Quantification of GRANZYME B labeling: Given the very small granule pattern of GRANZYME B labeling, the immunostained sections were scanned at X40 magnification and individual labeled cells were numbered. All the fields of the paraffin sections were assessed and the number of positive cells was counted in the infarct and peri-infarct areas at two different stages (≤7 days; >7 days).

**Population of patients with acute MI**. The population selection methods of the French registry of Acute ST-elevation and non-ST-elevation Myocardial Infarction (FAST-MI) have been described in detail in previous publications[42,43]. Briefly, all patients ≥18 years of age were included in the registry if they had elevated serum markers of myocardial necrosis higher than twice the upper limit of normal for creatine kinase, creatine kinase-MB, or elevated troponins, and either symptoms compatible with acute MI and/or electrocardiographic changes on at least two contiguous leads with pathologic Q waves (≥0.04 s) and/or persisting ST elevation or depression >0.1 mV. The time from symptom onset to intensive care unit admission had to be <48 h. Patients were managed according to usual practice; treatment was not affected by participation in the registry. Of the 374 centers in France that treated patients with acute MI at that time, 223 (60%) participated in the registry. Among these, 100 centers recruited 1029 patients who contributed to a serum bank. For the present study, 1046 samples were available for GRANZYME B measurement. Their baseline characteristics were comparable to the overall population of the registry. More than 99% of patients were Caucasians. Follow-up data were collected by contacting the patients' physicians, the patients themselves, or their family, and registry offices of their birthplace. Data obtained from 1-year follow-up were >99% complete. We have complied with all relevant ethical regulations. Written informed consent was provided by each patient. The study was approved by the Committee for the Protection of Human Subjects in Biomedical Research of Saint Antoine University Hospital (Ethical Committee) and the data file was declared to the Commission Nationale Informatique et Liberté (ClinicalTrials.gov NCT01237418). Human Granzyme B analysis was carried out using the Granzyme B Human ELISA Kit (Invitrogen, ThermoFisher), following the manufacturer's instructions.

**Statistical analysis**. Values are expressed as mean ± SEM. Differences between values were examined using the non-parametric Mann-Whitney test or Kruskal-Wallis test and were considered significant at $P < 0.05$. Shapiro-Wilk test was used for assessment of normality. Kaplan-Meier survival curves were constructed and analyzed using log-rank (Mantel-Cox) test. An outcome event was defined as all-cause death during the 1-year follow-up period. The primary endpoint was defined as all-cause death and was adjudicated by a committee whose members were unaware of patients' medications and blood measurements. Continuous variables are described as mean ± SD or median, Q1, Q3, and categorical variables as frequencies or percentages. Baseline demographic and clinical characteristics, treatment factors, and therapeutic management during hospitalization were compared among patients inferior or superior to the Granzyme B median level (8.9 pg/mL) using chi-square or Fisher's exact tests for discrete variables and by unpaired $t$-tests or Wilcoxon sign-rank tests for continuous variables. Survival curves according to the GRANZYME B median level are estimated using the Kaplan-Meier estimator. We used a multivariable Cox proportional-hazards model to assess the independent prognostic value of variables with the primary endpoint during the 1-year follow-up period. The multivariable model comprised sex, age, body mass index, current smoking, family history of coronary disease, history of hypertension, hypercholesterolemia, previous MI, previous stroke or transient ischemic attack (TIA), previous cancer, CPK peak, heart failure, renal failure, diabetes, Killip class, left ventricular ejection fraction, STEMI or reperfusion, and hospital management (including reperfusion therapy, coronary artery bypass surgery, statins, beta

blockers, clopidogrel, diuretics, low-molecular-weight heparin, GPIIb/IIIa inhibitors) (See Supplemental Table 3). Results are expressed as hazard ratios for Cox models with 95% confidence intervals (CIs). All statistical tests were two-sided and performed using SAS software version 9.4.

**Reporting summary**. Further information on research design is available in the Nature Research Reporting Summary linked to this article.

## Data availability
The data that support the findings of this study are available from the corresponding author on reasonable request. Source data are provided with this paper.

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

## Acknowledgements

This work was supported by Inserm, ANR (ANR-16-CE14-0016 to H.A.O., P.B., T.S., and J-S.S.), la Fondation Lefoulon-Delalande (Y.S.Z.), Inserm Transfert (I.S.Z., A.B., B.G., and H.A.O.). We are indebted to Andrew Lichtman and Ralf Dressel for providing CMy-mOva mice. We thank Carmen Marchiol and Gilles Renault (Plateforme Imageries du Vivant Institut Cochin – INSERM 1016) for their help in ultrasound evaluation of cardiac function. We are also indebted to the patients who accepted to participate and to all physicians who took care of them. We acknowledge the help of ICTA (Fontaine-lès-Dijon, France) and the clinical research team of Unité de Recherche Clinique de l'Est Parisien (URCEST, Assistance Publique des Hôpitaux de Paris), Vincent Bataille, Ph.D. (ADIMEP, Toulouse) for data collection and data management. Our gratitude also to the French Society of Cardiology for sponsorship and to Elodie Drouet, M.Sc., who supervised patient follow-up.

## Author contributions

I.S.Z., J.L., I.Z., M.V., R.C., M.E., J.B.S., C.H., C.D., J.S.H., C.C., P.B., V.D., Y.Z., M.B., A.B., and L.S. performed the experiments, acquired and interpreted the data. P.B. performed and interpreted the ultrasound studies. T.S. and N.D. were responsible for the FAST-MI cohort and interpreted the statistical data. J.V., E.D., J.C.S., H.B., and P.G. contributed to data acquisition and analysis. I.S.Z. performed the biomarkers measurements. M.C. performed the statistical analysis on the human data. P.B. analyzed and interpreted staining on human heart tissues. A.T., Z.M., and B.G. contributed to study design and data interpretation. A.T. and Z.M. reviewed the manuscript for important intellectual content. C.T. provided the CD8 mAb. I.S.Z., J-S.S., and H.A.O. designed the experiments, analyzed and interpreted the data. J.-S.S. and H.A.O. wrote the manuscript.

## Competing interests

H. Ait-Oufella applied a patent on the protective effect of CD8 depletion and Granzyme B blockade after acute myocardial infarction. The other authors declare no competing interests.
