## [Peer Review File · Nature Communications]

REVIEWER COMMENTS

Reviewer #1 (Remarks to the Author):

I appreciated the authors response and new figures to address my points 1-2. Especially the addition of Collagen 1 and 3 was very helpful. To point 3 authors showed that the population that is decreased after CD8 T cells depletion is Lychi inflammatory myeloid cells (I wish another macrophages then F4/80 was included). It is unclear from the supplemental figure 24 if the CD206 upregulation continues in CD8 depleted group and stops in controls. The figure showing differences in "M2 macrophages" is not persuasive since the control group on day 7 has no SD. To the point 4, I still believe that both sexes should have been included in the mice experiments, especially given the fact that there are differences between the results from the murine and porcine experiments and one un-accounted variable is sex (mice- males, pigs-females).

Reviewer #2 (Remarks to the Author):

The authors have addressed most of my concerns. New experiments and revisions have significantly strengthened the manuscript. I have no further recommendations.

Reviewer #3 (Remarks to the Author):

The authors have done an admirable job addressing my comments. They have strengthened their data in the pig model by further depleting CD8 T cells.

I have only one comments about the identification and quantification of macrophages which should be addressed.

In Supp Fig 17, the gating shows CD45+ cell in the heart, stratified by CD11b vs F4/80. However, I see virtually no F4/80+ macrophages in either group. I also see no control (uninfarcted group). The quantified data below demonstrates huge numbers (1000 cells/mg) of tissue. It is not clear where these data are coming from as the labeling appears to have not worked. F4/80 typically works very poorly by flow cytometry in the heart. The authors should use CD64 in the future.

This is linked to figure S17, which shows all CD11b+ cells, which the authors split into either Ly6g+ neutrophils or Ly6g- monocytes - yet this gate should contain macrophages. These issues need to be clarified.

Reviewer #4 (Remarks to the Author):

The authors have adequately responded to several of my previous concerns. Others remain to be addressed.

Major comments.

-MI model in mice. The authors state their outstanding experience in coronary ligation but my concern is about the experiments reported in this manuscript not the already published ones. It would be necessary to provide some data regarding this experiment regarding infarct size achieved by this operator.

-Timing of Antibody administration. To have comparable results between CD4 and Cd8 depletion the two antibodies should be administered at the same time.

-Figure 1c. An Isotype control staining should be provided for the actual myocardial staining. The ROI in fig 1 is not exactly matching the corresponded magnified area.

-Fig 2C. Isotype control staining should be included for the actual myocardial staining.

-Pig studies

In the pig model used the balloon was inserted surgically and inflated 14 days later. The majority of current pig studies achieve a precise coronary occlusion using a percutaneous balloon insertion and inflation under fluoroscopy for the required time. This approach is less invasive for the animals, more humane and it avoids the major immune perturbations of a surgical positioning. There is no evidence that a surgical intervention does not affect immune responses at 14 days.

The lack of follow up imaging at different time points remains a limitation of this study.

-Human studies

It would be valuable to compare with the human data to show GZMB level data in mice and pigs at least using a western blot or a proteomic approach.

Responses to reviewers

Reviewer #1:

I appreciated the authors response and new figures to address my points 1-2. Especially the addition of Collagen 1 and 3 was very helpful.

Response: We would like to thank the reviewer for his/her comments that helped to improve the manuscript

To point 3 authors showed that the population that is decreased after CD8 T cells depletion is Lychi inflammatory myeloid cells (I wish another macrophages then F4/80 was included).

*Response: We did not find any significant decrease either in *Lyc6Chigh* monocytes (CD45+CD11b+Ly6G-F4/80-Ly6Chigh) or in total macrophages (CD45+CD11b+Ly6G-F4/80+) in CD8 depleted animals. With all due respect, in our hands, F4/80 is a good and reliable marker to identify macrophages in ischemic heart tissue.*

It is unclear from the supplemental figure 24 if the CD206 upregulation continues in CD8 depleted group and stops in controls. The figure showing differences in “M2 macrophages” is not persuasive since the control group on day 7 has no SD.

Response: We analyzed total macrophage population in the ischemic heart tissue at day 5 after MI. We did not observe any difference between CTR and CD8 depleted groups (Supplementary fig. 20). We also analyzed the amount of cardiac M2-like reparative macrophages (CD45+Ly6G-F4/80+CD206+) at day 1, 3 and 7 after MI. We found higher number of M2-like reparative macrophages in CD8 depleted group, compared to control group (Supplementary figure 26). The SD bar at Day7 was indeed missing in the previous version of the figure, we do apologize for this omission. This has been corrected in the revised version of the manuscript.

Finally, we analyzed mRNA expression of isolated macrophages from infarcted heart at day 5 after MI. We observed a shift in gene expression profile toward a less inflammatory phenotype (Supplementary figure 25). Altogether, our findings using different experimental approaches strongly suggest a shift in the cardiac macrophage phenotype toward a M2-like reparative profile in CD8 depleted mice after MI.

To the point 4, I still believe that both sexes should have been included in the mice experiments, especially given the fact that there are differences between the results from the murine and porcine experiments and one un-accounted variable is sex (mice- males, pigs-females).

Response: In a substantial effort to address this reviewer's concern, we carried out an additional set of experiments in 10-week old female C57Bl6 mice using the same experimental protocol as the one used in males. We confirmed that CD8 T cell depletion limited deleterious post-ischemic cardiac remodeling in females (New Supplementary figure 13).

Reviewer #2 (Remarks to the Author):

The authors have addressed most of my concerns. New experiments and revisions have significantly strengthened the manuscript. I have no further recommendations.

Response: We would like to thank the reviewer for his/her encouraging comments that helped to improve the manuscript

Reviewer #3 (Remarks to the Author):

The authors have done an admirable job addressing my comments. They have strengthened their data in the pig model by further depleting CD8 T cells.

Response: We would like to thank the reviewer for his/her comments that helped to improve the manuscript

I have only one comments about the identification and quantification of macrophages which should be addressed.

In Supp Fig 17, the gating shows CD45+ cell in the heart, stratified by CD11b vs F4/80. However, I see virtually no F4/80+ macrophages in either group. I also see no control (uninfarcted group). The quantified data below demonstrates huge numbers (1000 cells/mg) of tissue. It is not clear where these data are coming from as the labeling appears to have not worked. F4/80 typically works very poorly by flow cytometry in the heart. The authors should use CD64 in the future.

Response: We would like to thank the reviewer for his/her comment but in our hands, F4/80 is a reliable marker for macrophages. Nevertheless, we take good note of the sound advice of this reviewer The flow picture has been replaced by a new one (Supplemental figure 20).

We did not include control un-infarcted group because we aimed to specifically investigate the impact of CD8 depletion on macrophage content in the ischemic heart tissue. As a consequence, our strategy was to compare the effect of isotype vs anti-CD8 treatment administration.

This is linked to figure S17, which shows all CD11b+ cells, which the authors split into either Ly6g+ neutrophils or Ly6g- monocytes - yet this gate should contain macrophages. These issues need to be clarified.

Response: Based on the reviewer's comment we double checked our data and gating strategy. Indeed, F4/80+ cells were excluded to identify monocyte populations (CD45+CD11b+Ly6G-F4/80-Ly6Ghigh/low). Supplementary Fig.19 has been clarified accordingly.

Reviewer #4 (Remarks to the Author):

The authors have adequately responded to several of my previous concerns. Others remain to be addressed.

Response: We would like to thank the reviewer for his/her comments that helped to improve the manuscript

Major comments.

-MI model in mice. The authors state their outstanding experience in coronary ligation but my concern is about the experiments reported in this manuscript not the already published ones. It would be necessary to provide some data regarding this experiment regarding infarct size achieved by this operator.

Response: We apologize to this reviewer if we were presumptuous in our previous answer, the idea was simply to underline the mastery of this experimental approach that we have acquired over time. We have performed additional experiments with TTC staining at day 3 to analyze infarct size. As depicted in Figure 24, variability in each group was very low, confirming good reproducibility of coronary ligation in our experimental setting.

-Timing of Antibody administration. To have comparable results between CD4 and CD8 depletion the two antibodies should be administered at the same time.

Response: We do apologize for the misunderstanding regarding the experimental design in figure 1 I-J-K. The idea here was to analyze the role of CD4+ T cells on the recruitment/trafficking of CD8+ T cells We have therefore injected anti-CD4 depleting antibody to obtain full CD4+ T cell depletion during the coronary ligation and subsequently we analyzed CD8+ T cell population in different compartment after MI. We found that CD4 depletion limited CD8 mobilization from the spleen to the blood and ultimately to the ischemic heart.

-Figure 1c/2c. An Isotype control staining should be provided for the actual myocardial staining. The ROI in fig 1 is not exactly matching the corresponded magnified area.

Response: As suggested, an isotype control staining has been done and added in supplementary Fig 2. ROI has been modified accordingly.

-Pig studies

in the pig model used the balloon was inserted surgically and inflated 14 days later. The majority of current pig studies achieve a precise coronary occlusion using a percutaneous balloon insertion and inflation under fluoroscopy for the required time. This approach is less invasive for the animals, more humane and it avoids the major immune perturbations of a surgical positioning. There is no evidence that a surgical intervention does not affect immune responses at 14 days. The lack of follow up imaging at different time points remains a limitation of this study.

Response: We do agree with the reviewer that endovascular coronary occlusion is less invasive than surgical intervention. This endovascular procedure will be developed in our lab in the future.

-Human studies

It would be valuable to compare with the human data to show Granzyme B level data in mice and pigs at least using a western blot or a proteomic approach.

Response: As suggested, we measured by ELISA Granzyme B plasma levels in CTR (N=4, 2 samples are not available) and CD8 depleted pigs (N=5) at different time point after the onset of MI. Granzyme B was detected at day 1 and day 3 after MI. Granzyme B levels were very low at day 14. We found a significant decrease in Granzyme B plasma levels in CD8 depleted pigs. This result has been added in supplementary fig.38.

REVIEWERS' COMMENTS

Reviewer #1 (Remarks to the Author):

I appreciate the authors effort to clarify the remaining issues, especially the supplemental figure 13 has cleared the sex differences issue. The manuscript is also improved by clarifying the myeloid cells gating strategy. I would only urge authors to truly add another macrophages marker (CD64) and use FMO of F4/80 gating in their future work. As mentioned by me and reviewer #3, F4/80 is not considered reliable macrophages marker in the heart. This issue however should not hold this manuscript in publication at this point.

Reviewer #3 (Remarks to the Author):

I have no further questions - the authors have addressed all my concerns.

Reviewer #4 (Remarks to the Author):

The authors have adequately responded to my previous comments.

One point remains regarding the pig studies as the surgical approach could have induced immune perturbations that cannot be excluded. The authors should acknowledge this clearly in the discussion.

Responses to the reviewers

Reviewer #1:

I appreciate the authors effort to clarify the remaining issues, especially the supplemental figure 13 has cleared the sex differences issue. The manuscript is also improved by clarifying the myeloid cells gating strategy. I would only urge authors to truly add another macrophages marker (CD64) and use FMO of F4/80 gating in their future work. As mentioned by me and reviewer #3, F4/80 is not considered reliable macrophages marker in the heart. This issue however should not hold this manuscript in publication at this point.

Response: we would like to thank the reviewer for his comments that helped to improve the manuscript

Reviewer #3 (Remarks to the Author):

I have no further questions - the authors have addressed all my concerns.

Response: we would like to thank the reviewer for his comments that helped to improve the manuscript

Reviewer #4 (Remarks to the Author):

The authors have adequately responded to my previous comments.

One point remains regarding the pig studies as the surgical approach could have induced immune perturbations that cannot be excluded. The authors should acknowledge this clearly in the discussion.

Responses: we would like to thank the reviewer for his comments that helped to improve the manuscript. The limitation has been inserted a the end of the discussion Section.